# Neural population dynamics of computing with synaptic modulations

**Kyle Aitken\*, Stefan Mihalas**

Allen Institute, MindScope Program, Seattle, United States

**Abstract** In addition to long-timescale rewiring, synapses in the brain are subject to significant modulation that occurs at faster timescales that endow the brain with additional means of processing information. Despite this, models of the brain like recurrent neural networks (RNNs) often have their weights frozen after training, relying on an internal state stored in neuronal activity to hold task-relevant information. In this work, we study the computational potential and resulting dynamics of a network that relies solely on synapse modulation during inference to process task-relevant information, the multi-plasticity network (MPN). Since the MPN has no recurrent connections, this allows us to study the computational capabilities and dynamical behavior contributed by synapses modulations alone. The generality of the MPN allows for our results to apply to synaptic modulation mechanisms ranging from short-term synaptic plasticity (STSP) to slower modulations such as spike-time dependent plasticity (STDP). We thoroughly examine the neural population dynamics of the MPN trained on integration-based tasks and compare it to known RNN dynamics, finding the two to have fundamentally different attractor structure. We find said differences in dynamics allow the MPN to outperform its RNN counterparts on several neuroscience-relevant tests. Training the MPN across a battery of neuroscience tasks, we find its computational capabilities in such settings is comparable to networks that compute with recurrent connections. Altogether, we believe this work demonstrates the computational possibilities of computing with synaptic modulations and highlights important motifs of these computations so that they can be identified in brain-like systems.

**\*For correspondence:**
kyle.aitken@alleninstitute.org

**Competing interest:** The authors declare that no competing interests exist.

## Editor's evaluation

The study shows that fast and transient modifications of the synaptic efficacies, alone, can support the storage and processing of information over time. Convincing evidence is provided by showing that feed-forward networks, when equipped with such short-term synaptic modulations, perform a wide variety of tasks at a performance level comparable with that of recurrent networks. The results of the study are valuable to both neuroscientists and researchers in machine learning.

## Introduction

The brain's synapses constantly change in response to information under several distinct biological mechanisms (*Love, 2003*; *Hebb, 2005*; *Bailey and Kandel, 1993*; *Markram et al., 1997*; *Bi and Poo, 1998*; *Stevens and Wang, 1995*; *Markram and Tsodyks, 1996*). These changes can serve significantly different purposes and occur at drastically different timescales. Such mechanisms include synaptic rewiring, which modifies the topology of connections between neurons in our brain and can be as fast as minutes to hours. Rewiring is assumed to be the basis of long-term memory that can last a lifetime (*Bailey and Kandel, 1993*). At faster timescales, individual synapses can have their strength modified (*Markram et al., 1997*; *Bi and Poo, 1998*; *Stevens and Wang, 1995*; *Markram and Tsodyks, 1996*). These changes can occur over a spectrum of timescales and can be intrinsically transient (*Stevens and Wang, 1995*; *Markram and Tsodyks, 1996*). Though such mechanisms may not immediately lead to structural changes, they are thought to be vital to the brain's function. For

example, short-term synaptic plasticity (STSP) can affect synaptic strength on timescales less than a second, with such effects mainly presynaptic-dependent (*Stevens and Wang, 1995*; *Tsodyks and Markram, 1997*). At slower timescales, long-term potentiation (LTP) can have effects over minutes to hours or longer, with the early phase being dependent on local signals and the late phase including a more complex dependence on protein synthesis (*Baltaci et al., 2019*). Also on the slower end, spike-time-dependent plasticity (STDP) adjusts the strengths of connections based on the relative timing of pre- and postsynaptic spikes (*Markram et al., 1997*; *Bi and Poo, 1998*; *McFarlan et al., 2023*).

In this work, we investigate a new type of artificial neural network (ANN) that uses biologically motivated synaptic modulations to process short-term sequential information. The *multi-plasticity network* (MPN) learns using two complementary plasticity mechanisms: (1) long-term synaptic rewiring via standard supervised ANN training and (2) simple synaptic modulations that operate at faster timescales. Unlike many other neural network models with synaptic dynamics (*Tsodyks et al., 1998*; *Mongillo et al., 2008*; *Lundqvist et al., 2011*; *Barak and Tsodyks, 2014*; *Orhan and Ma, 2019*; *Ballintyn et al., 2019*; *Masse et al., 2019*), *the MPN has no recurrent synaptic connections*, and thus can only rely on modulations of synaptic strengths to pass short-term information across time. Although both recurrent connections and synaptic modulation are present in the brain, it can be difficult to isolate how each of these affects temporal computation. The MPN thus allows for an in-depth study of the computational power of synaptic modulation alone and how the dynamics behind said computations may differ from networks that rely on recurrence. Having established how modulations alone compute, we believe it will be easier to disentangle synaptic computations from brain-like networks that may compute using a combination of recurrent connections, synaptic dynamics, neuronal dynamics, etc.

Biologically, the modulations in the MPN represent a general synapse-specific change of strength on shorter timescales than the structural changes, the latter of which are represented by weight adjustment via backpropagation. We separately consider two forms of modulation mechanisms, one of which is dependent on *both* the pre- and postsynaptic firing rates and a second that only depends on presynaptic rates. The first of these rules is primarily envisioned as coming from associative forms of plasticity that depend on both pre- and postsynaptic neuron activity (*Markram et al., 1997*; *Bi and Poo, 1998*; *McFarlan et al., 2023*). Meanwhile, the second type of modulation models presynaptic-dependent STSP (*Mongillo et al., 2008*; *Zucker and Regehr, 2002*). While both these mechanisms can arise from distinct biological mechanisms and can span timescales of many orders of magnitude, the MPN uses simplified dynamics to keep the effects of synaptic modulations and our subsequent results as general as possible. It is important to note that in the MPN, as in the brain, the mechanisms that represent synaptic modulations and rewiring are not independent of one another – changes in one affect the operation of the other and vice versa.

To understand the role of synaptic modulations in computing and how they can change neuronal dynamics, throughout this work we contrast the MPN with recurrent neural networks (RNNs), whose synapses/weights remain fixed after a training period. RNNs store temporal, task-relevant information in transient internal neural activity using recurrent connections and have found widespread success in modeling parts of our brain (*Cannon et al., 1983*; *Ben-Yishai et al., 1995*; *Seung, 1996*; *Zhang, 1996*; *Ermentrout, 1998*; *Stringer et al., 2002*; *Xie et al., 2002*; *Fuhs and Touretzky, 2006*; *Burak and Fiete, 2009*). Although RNNs model the brain's significant recurrent connections, the weights in these networks neglect the role transient synaptic dynamics can have in adjusting synaptic strengths and processing information.

Considerable progress has been made in analyzing brain-like RNNs as population-level dynamical systems, a framework known as *neural population dynamics* (*Vyas et al., 2020*). Such studies have revealed a striking universality of the underlying computational scaffold across different types of RNNs and tasks (*Maheswaranathan et al., 2019b*). To elucidate how computation through synaptic modulations affect neural population behavior, we thoroughly characterize the MPN's low-dimensional behavior in the neural population dynamics framework (*Vyas et al., 2020*). Using a novel approach of analyzing the synapse population behavior, we find the MPN computes using completely different dynamics than its RNN counterparts. We then explore the potential benefits behind its distinct dynamics on several neuroscience-relevant tasks.

## Contributions
The primary contributions and findings of this work are as follows:

- We elucidate the neural population dynamics of the MPN trained on integration-based tasks and show it operates with qualitatively different dynamics and attractor structure than RNNs. We support this with analytical approximations of said dynamics.
- We show how the MPN's synaptic modulations allow it to store and update information in its state space using a task-independent, single point-like attractor, with dynamics slower than task-relevant timescales.
- Despite its simple attractor structure, for integration-based tasks, we show the MPN performs at level comparable or exceeding RNNs on several neuroscience-relevant measures.
- The MPN is shown to have dynamics that make it a more effective reservoir, less susceptible to catastrophic forgetting, and more flexible to taking in new information than RNN counterparts.
- We show the MPN is capable of learning more complex tasks, including contextual integration, continuous integration, and 19 neuroscience tasks in the NeuroGym package (*Molano-Mazon et al., 2022*). For a subset of tasks, we elucidate the changes in dynamics that allow the network to solve them.

## Related work

Networks with synaptic dynamics have been investigated previously (*Tsodyks et al., 1998*; *Mongillo et al., 2008*; *Sugase-Miyamoto et al., 2008*; *Lundqvist et al., 2011*; *Barak and Tsodyks, 2014*; *Orhan and Ma, 2019*; *Ballintyn et al., 2019*; *Masse et al., 2019*; *Hu et al., 2021*; *Tyulmankov et al., 2022*; *Tyulmankov et al., 2022*; *Rodriguez et al., 2022*). As we mention above, many of these works investigate networks with both synaptic dynamics and recurrence (*Tsodyks et al., 1998*; *Mongillo et al., 2008*; *Lundqvist et al., 2011*; *Barak and Tsodyks, 2014*; *Orhan and Ma, 2019*; *Ballintyn et al., 2019*; *Masse et al., 2019*), whereas here we are interested in investigating the computational capabilities and dynamical behavior of computing with synapse modulations alone. Unlike previous works that examine computation solely through synaptic changes, the MPN's modulations occur at all times and do not require a special signal to activate their change (*Sugase-Miyamoto et al., 2008*). The networks examined in this work are most similar to the recently introduced 'HebbFF' (*Tyulmankov et al., 2022*) and 'STPN' (*Rodriguez et al., 2022*) that also examine computation through continuously updated synaptic modulations. Our work differs from these studies in that we focus on elucidating the neural population dynamics of such networks, contrasting them to known RNN dynamics, and show why this difference in dynamics may be beneficial in certain neuroscience-relevant settings. Additionally, the MPN uses a multiplicative modulation mechanism rather than the additive modulation of these two works, which in some settings we investigate yields significant performance differences. The exact form of the synaptic modulation updates were originally inspired by 'fast weights' used in machine learning for flexible learning (*Ba et al., 2016*). However, in the MPN, both plasticity rules apply to the same weights rather than different ones, making it more biologically realistic.

This work largely focuses on understanding computation through a *neural population dynamics*-like analysis (*Vyas et al., 2020*). In particular, we focus on the dynamics of networks trained on integration-based tasks, that have previously been studied in RNNs (*Maheswaranathan et al., 2019b*; *Maheswaranathan et al., 2019a*; *Maheswaranathan and Sussillo, 2020*; *Aitken et al., 2020*). These studies have demonstrated a degree of universality of the underlying computational structure across different types of tasks and RNNs (*Maheswaranathan et al., 2019b*). Due to the MPN's dynamic weights, its operation is fundamentally different than said recurrent networks.

## Setup

Throughout this work, we primarily investigate the dynamics of the MPN on tasks that require an integration of information over time. To correctly respond to said task, the network is required to both store and update its internal state as well as compare several distinct items in its memory. All tasks in this work consist of a discrete sequence of vector inputs, $\mathbf{x}_t$ for $t = 1, 2, \ldots, T$. For the tasks we consider presently, at time $T$ the network is queried by a 'go signal' for an output, for which the correct response can depend on information from the entire input sequence. Throughout this paper, we denote vectors using lowercase bold letters, matrices by uppercase bold letters, and scalars using standard (not-bold) letters. The input, hidden, and output layers of the networks we study have $d$, $n$, and $N$ neurons, respectively.

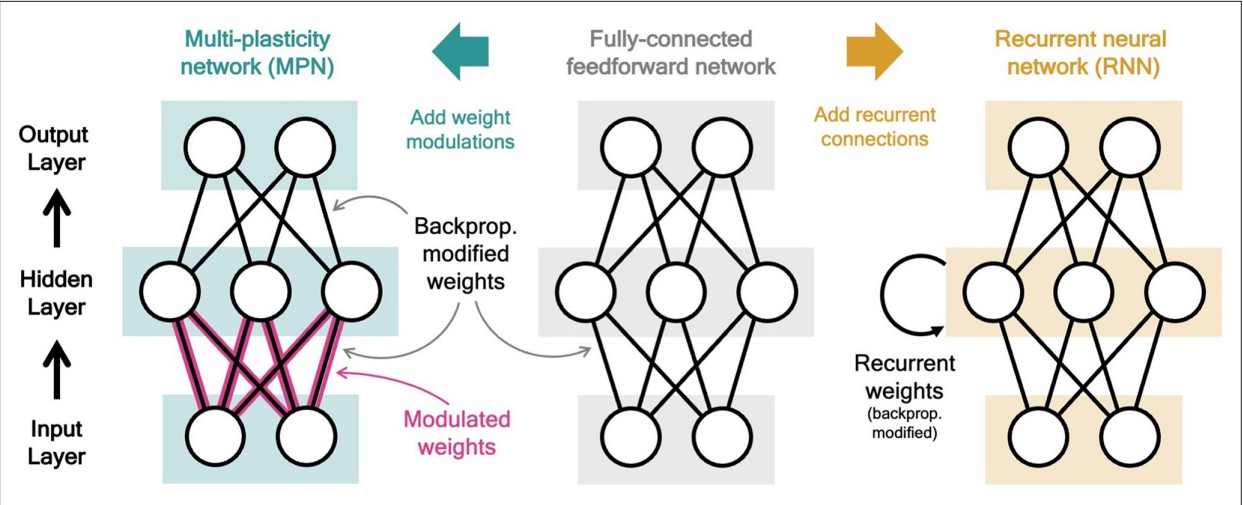

**Figure 1.** Two neural network computational mechanisms: synaptic modulations and recurrence. Throughout this figure, neurons are represented as white circles, the black lines between neurons represent regular feedforward weights that are modified during training through gradient descent/ backpropagation. From bottom to top are the input, hidden, and output layers, respectively. (Middle) A two-layer, fully connected, feedforward neural network. (Left) Schematic of the MPN. Here, the pink and black lines (between the input and hidden layer) represent weights that are modified by *both* backpropagation (during training) and the synapse modulation matrix (during an input sequence), see *Equation 1*. (Right) Schematic of the Vanilla RNN. In addition to regular feedforward weights between layers, the RNN has (fully connected) weights between its hidden layer from one time step to the next, see *Equation 3*.

## Multi-plasticity network

The *multi-plasticity network (MPN)* is an artificial neural network consisting of input, hidden, and output layers of neurons. It is identical to a fully-connected, two-layer, feedforward network (*Figure 1*, middle), with one major exception: the weights connecting the input and hidden layer are modified by the time-dependent *synapse modulation (SM) matrix*, $\mathbf{M}$ (*Figure 1*, left). The expression for the hidden layer activity at time step $t$ is

$$\mathbf{h}_t \quad = \tanh\left(\left(\mathbf{M}_{t-1} \odot \mathbf{W}_{\text{inp}}\right)\mathbf{x}_t + \mathbf{W}_{\text{inp}}\mathbf{x}_t\right) \tag{1}$$

where $\mathbf{W}_{\text{inp}}$ is an $n$-by-$d$ weight matrix representing the network's synaptic strengths that is fixed after training, '$\odot$' denotes element-wise multiplication of the two matrices (the Hadamard product), and the $\tanh(\cdot)$ is applied element-wise. For each synaptic weight in $\mathbf{W}_{\text{inp}}$, a corresponding element of $\mathbf{M}_{t-1}$ multiplicatively modulates its strength. Note if $\mathbf{M}_{t-1} = \mathbf{0}$ the first term vanishes, so the $\mathbf{W}_{\text{inp}}$ are unmodified and the network simply functions as a fully connected feedforward network.

What allows the MPN to store and manipulate information as the input sequence is passed to the network is how the SM matrix, $\mathbf{M}_t$, changes over time. Throughout this work, we consider two distinct modulation update rules. The primary rule we investigate is dependent upon *both* the pre- and postsynaptic firing rates. An alternative update rule only depends upon the presynaptic firing rate. Respectively, the SM matrix updated for these two cases takes the form (*Hebb, 2005*; *Ba et al., 2016*; *Tyulmankov et al., 2022*),

$$pre.\&\,post.: \mathbf{M}_t = \lambda\mathbf{M}_{t-1} + \eta\mathbf{h}_t\mathbf{x}_t^T \tag{2a}$$

$$pre.\,only: M_t = \lambda\mathbf{M}_{t-1} + \eta\mathbf{1}\mathbf{x}_t^T/\sqrt{n}\,, \tag{2b}$$

where $\lambda$ and $\eta$ are parameters learned during training and $\mathbf{1}$ is the n-dimensional vector of all 1s. We allow for $-\infty < \eta < \infty$, so the size and sign of the modulations can be optimized during training. Additionally, $0 < \lambda < 1$, so the SM matrix exponentially decays at each time step, asymptotically returning to its $\mathbf{M} = \mathbf{0}$ baseline. For both rules, we define $\mathbf{M}_0 = \mathbf{0}$ at the start of each input sequence. Since the SM matrix is updated and passed forward at each time step, we will often refer to $\mathbf{M}_t$ as the *state* of said networks.

To distinguish networks with these two modulation rules, we will refer to networks with the presynaptic only rule as *MPNpre*, while we reserve *MPN* for networks with the pre- and postsynatpic update that we primarily investigate. For brevity, and since almost all results for the MPN generalize to the simplified update rule of the MPNpre, the main text will foremost focus on results for the MPN. Results for the MPNpre are discussed only briefly or given in the supplement.

As mentioned in the introduction, from a biological perspective the MPN's modulations represent a general associative plasticity such as STDP, whereas the presynaptic-dependent modulations of the MPNpre can represent STSP. The decay induced by $\lambda$ represents the return to baseline of the aforementioned processes, which all occur at a relatively slow speed to their onset (*Bertram et al., 1996*; *Zucker and Regehr, 2002*). To ensure the eventual decay of such modulations, unless otherwise stated, throughout this work we further limit $\lambda < \lambda^{\text{max}}$ with $\lambda^{\text{max}} = 0.95$. Additionally, we observe no major performance or dynamics difference for positive or negative $\eta$, so we do not distinguish the two throughout this work (Methods). We emphasize that the modulation mechanisms of the MPN and MPNpre could represent biological processes that occur at significantly different timescales, so although we train them on identical tasks the tasks themselves are assumed to occur at timescales that match the modulation mechanism of the corresponding network. Note that the modulation mechanisms are not independent of weight adjustment from backpropagation. Since the SM matrix is active during training, the network's weights that are being adjusted by backpropgation (see below) are experiencing modulations, and said modulations factor into how the weights are adjusted.

Lastly, the output of the MPN and MPNpre at time $T$ is determined by a fully-connected readout matrix, $\mathbf{y}_T = \mathbf{W}_{\text{RO}}\mathbf{h}_T$, where $\mathbf{W}_{\text{RO}}$ is an $N$-by-$n$ weight matrix adjusted during training. Throughout this work, we will view said readout matrix as $N$ distinct $n$-dimensional readout vectors, that is one for each output neuron.

## Recurrent neural networks

As discussed in the introduction, throughout this work we will compare the learned dynamics and performance of the MPN to artificial RNNs. The hidden layer activity for the simplest recurrent neural network, the *Vanilla RNN*, is

$$\mathbf{h}_t = \tanh\left(\mathbf{W}_{\text{rec}}\mathbf{h}_{t-1} + \mathbf{W}_{\text{inp}}\mathbf{x}_t + \mathbf{b}\right), \tag{3}$$

with $\mathbf{W}_{\text{rec}}$ the recurrent weights, an $n$-by-$n$ matrix that updates the hidden neurons from one time step to the next (*Figure 1*, right). We also consider a more sophisticated RNN structure, the *gated recurrent unit (GRU)*, that has additional gates to more precisely control the recurrent update of its hidden neurons (see Methods 5.2). In both these RNNs, information is stored and updated via the hidden neuron activity, so we will often refer to $\mathbf{h}_t$ as the RNNs' *hidden state* or just its *state*. The output of the RNNs is determined through a trained readout matrix in the same manner as the MPN above, i.e. $\mathbf{y}_T = \mathbf{W}_{\text{RO}}\mathbf{h}_T$.

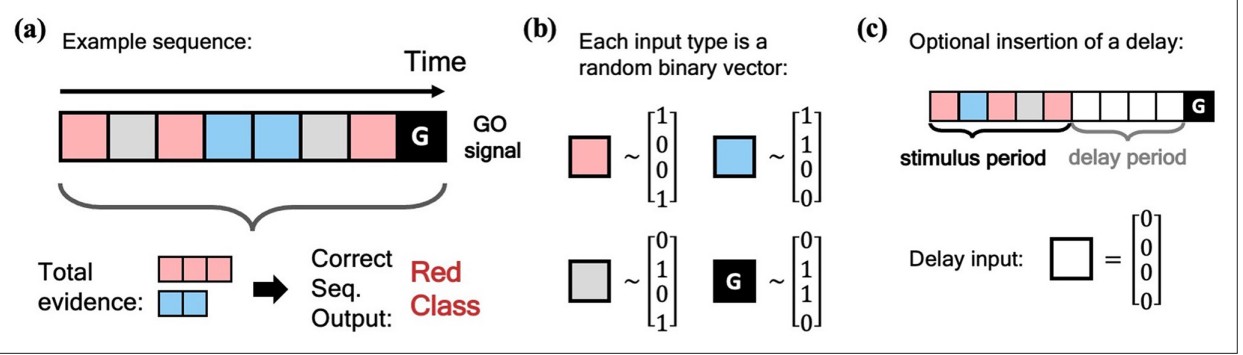

**Figure 2.** Schematic of simple integration task. (**a**) Example sequence of the two-class integration task where each box represents an input. Here and throughout this work, distinct classes are represented by different colors. In this case, red and blue. The red/blue boxes represent evidence for their respective classes, while the grey box represents an input that is evidence for neither class. At the end of the sequence is the 'go signal' that lets the network know an output is expected. The correct response for the sequence is the class with the most evidence; in the example shown, the red class. (**b**) Each possible input is mapped to a (normalized) random binary vector. (**c**) The integration task can be modified by the insertion of a 'delay period' between the stimulus period and the go signal. During the delay period, the network receives no input (other than noise).

## Training

The weights of the MPN, MPNpre, and RNNs will be trained using gradient descent/backpropagation through time, specifically ADAM (*Kingma and Ba, 2014*). All network weights are subject to L1 regularization to encourage sparse solutions (Methods 5.2). Cross-entropy loss is used as a measure of performance during training. Gaussian noise is added to all inputs of the networks we investigate.

## Results

## Network dynamics on a simple integration task

### Simple integration task

We begin our investigation of the MPN's dynamics by training it on a simple N-class (Through most of this work, the number of neurons in the output layer of our networks will always be equal to the number of classes in the task, so we use $N$ to denote both unless otherwise stated). integration task, inspired by previous works on RNN integration-dynamics (*Maheswaranathan et al., 2019a*; *Aitken et al., 2020*). In this task, the network will need to determine for which of the $N$ classes the input sequence contains the most evidence (*Figure 2a*). Each stimulus input, $\mathbf{x}_t$, can correspond to a discrete unit of evidence for one of the $N$ classes. We also allow inputs that are evidence for none of the classes. The final input, $\mathbf{x}_T$, will always be a special 'go signal' input that tells the network an output is expected. The network's output should be an integration of evidence over the entire input sequence, with an output activity that is largest from the neuron that corresponds to the class with the maximal accumulated evidence. (We omit sequences with two or more classes tied for the most evidence. See Methods 5.1 for additional details). Prior to adding noise, each possible input, including the go signal, is mapped to a random binary vector (*Figure 2b*). We will also investigate the effect of inserting a *delay period* between the stimulus period and the go signal, during which no input is passed to the network, other than noise (*Figure 2c*).

We find the MPN (and MPNpre) is capable of learning the above integration task to near perfect accuracy across a wide range of class counts, sequence lengths, and delay lengths. It is the goal of this section to illuminate the dynamics behind the trained MPN that allow it to solve such a task and compare them to more familiar RNN dynamics. Here, in the main text, we will explicitly explore the dynamics of a two-class integration task, generalizations to $N > 2$ classes are straightforward and are discussed in the Methods 5.4. We will start by considering the simplest case of integration without a delay period, revisiting the effects of delay afterwards.

Before we dive into the dynamics of the MPN, we give a quick recap of the known RNN dynamics on integration-based tasks.

### Review of RNN integration: attractor dynamics encodes accumulated evidence

Several studies, both on natural and artificial neural networks, have discovered that networks with recurrent connections develop attractor dynamics to solve integration-based tasks (*Maheswaranathan et al., 2019a*; *Maheswaranathan and Sussillo, 2020*; *Aitken et al., 2020*). Here, we specifically review the behavior of artificial RNNs on the aforementioned N-class integration tasks that share many qualitative features with experimental observations of natural neural networks. Note also the structure/dimensionality of the dynamics can depend on correlations between the various classes (*Aitken et al., 2020*), in this work we only investigate the case where the various classes are uncorrelated.

RNNs are capable of learning to solve the simple integration task at near-perfect accuracy and their dynamics are qualitatively the same across several architectures (*Maheswaranathan et al., 2019b*; *Aitken et al., 2020*). Discerning the network's behavior by looking at individual hidden neuron activity can be difficult (*Figure 3a*), and so it is useful to turn to a population-level analysis of the dynamics. When the number of hidden neurons is much larger than number of integration classes ($n \gg N$), the population activity of the trained RNN primarily exists in a low-dimensional subspace of approximate dimension $N - 1$ (*Aitken et al., 2020*). This is due to recurrent dynamics that create a task-dependent attractor manifold of approximate dimension $N - 1$, and the hidden activity often operates close to said attractor. (See Methods 5.4 for a more in-depth review of these results including how approximate dimensionality is determined). In the two-class case, the RNN will operate close to a finite length

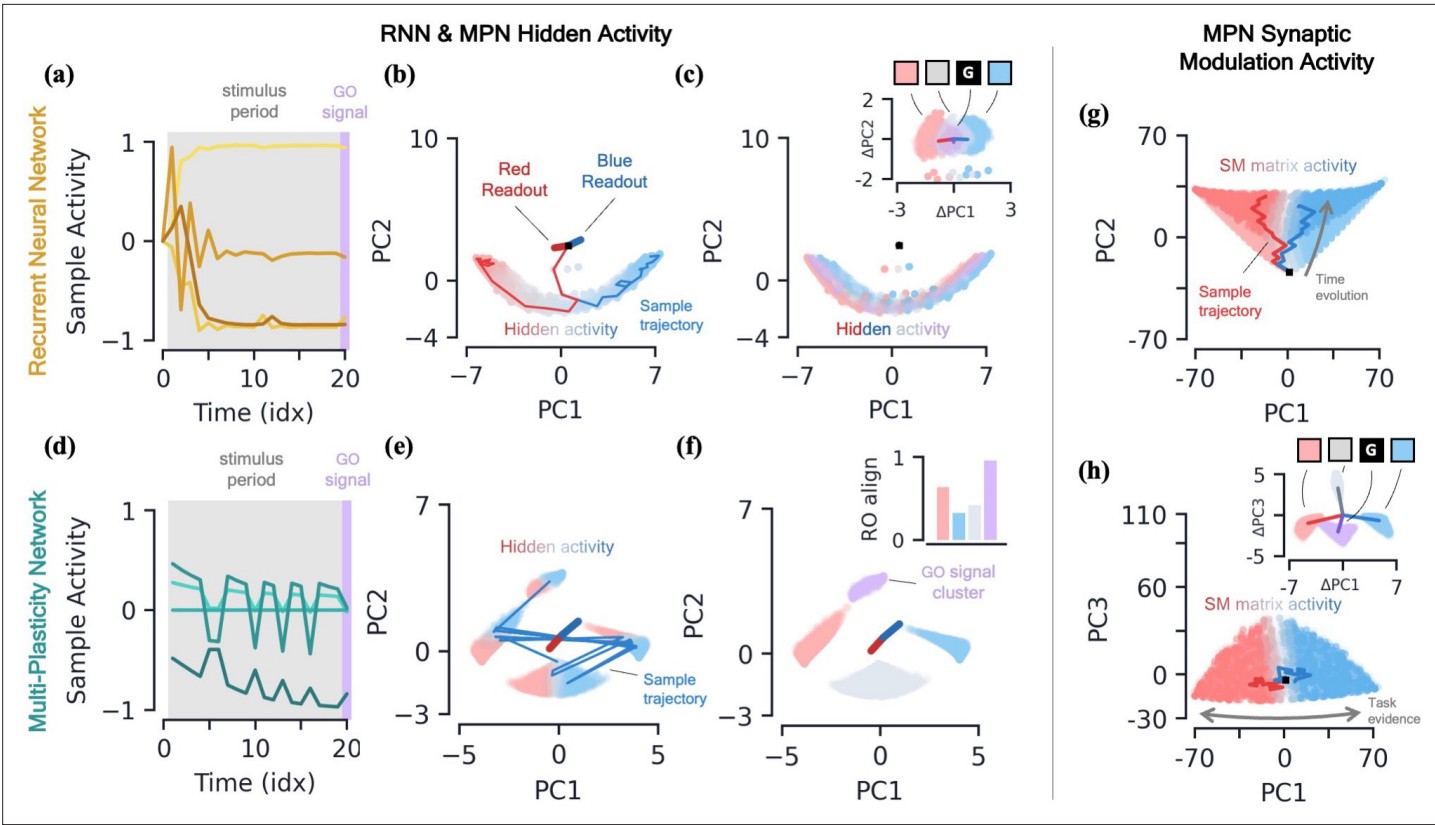

**Figure 3.** Two-class integration: comparison of multi-plasticity network and RNN dynamics. (**a-c**) Vanilla RNN hidden neuron dynamics, see **Figure 3—figure supplement 1** for GRU. (**a**) Hidden layer neural activity, $\mathbf{h}_t$, for four sample neurons of the RNN as a function of sequence time (in units of sequence index). The shaded grey region represents the stimulus period during which information should be integrated across time and the thin purple-shaded region representing the response to the go signal. (**b**) Hidden neuron activity, collected over 1000 input sequences, projected into their top two PCA components, colored by relative accumulated evidence between red/blue classes at time $t$ (Methods 5.5). Also shown are PCA projections of sample trajectories (thin lines, colored by class), the red/blue class readout vector (thick lines), and the initial state (black square). (**c**) Same as (**b**), with $\mathbf{h}_t$ now colored by input at the present time step, $\mathbf{x}_t$ (four possibilities, see inset). The inset shows the PCA projection of $\mathbf{h}_t - \mathbf{h}_{t-1}$ as a function of the present input, $\mathbf{x}_t$, with the dark lines showing the average for each of the four inputs. [d-f] *MPN hidden neuron dynamics, see **Figure 3—figure supplement 1** for MPNpre.* (**d**) Same as (**a**). (**e**) Same as (**b**). (**f**) Same as (**c**), except for inset. The inset now shows the alignment of each input-cluster with the readout vectors (Methods 5.5). [g-h] *MPN synaptic modulation dynamics.* (**g**) Same as (**b**), but instead of hidden neuron activity, the PCA projection of the SM matrices, $\mathbf{M}_t$, collected over 1000 input sequences. Final $\mathbf{M}_t$ are colored slightly darker for clarity. (**h**) Same as (**g**), with a different $y$-axis. The inset is the same as that shown in (**b**), but for $\mathbf{M}_t - \mathbf{M}_{t-1}$.

The online version of this article includes the following figure supplement(s) for figure 3:

**Figure supplement 1.** Two-class integration: additional network dynamics.

**Figure supplement 2.** Two-class integration: additive dynamics and modulation bounds.

**Figure supplement 3.** 3-class integration: MPN, Vanilla RNN, and GRU dynamics.

line attractor. The low-dimensionality of hidden activity allows for an intuitive visualization of the dynamics using a two-dimensional PCA projection (**Figure 3b**). From the sample trajectories, we see the network's hidden activity starts slightly offset from the line attractor before quickly falling towards its center. As evidence for one class over the other builds, *the hidden activity encodes accumulated evidence by moving along the one-dimensional attractor* (**Figure 3b**). The two readout vectors are roughly aligned with the two ends of the line, so the further the final hidden activity, $\mathbf{h}_T$, is toward one side of the attractor, the higher that class's corresponding output and thus the RNN correctly identifies the class with the most evidence. For later reference, we note that the hidden activity of the trained RNN is not highly dependent upon the input of the present time step (**Figure 3c**), but instead it is the *change* in the hidden activity from one time step to the next, $\mathbf{h}_t - \mathbf{h}_{t-1}$, that are highly input-dependent (**Figure 3c**, inset). For the Vanilla RNN (GRU), we find $0.53 \pm 0.01$ $(0.88 \pm 0.01)$ of the hidden

activity variance to be explained by the accumulated evidence and only $0.19 \pm 0.01$ ($0.29 \pm 0.01$) to be explained by the present input to the network (mean±s.e., Methods 5.4).

## MPN hidden activity encodes inputs, not so much accumulated evidence

We now turn to analyzing the hidden activity of the trained MPNs in the same manner that was done for the RNNs. The MPN trained on a two-class integration task appears to have significantly more sporadic activity in the individual components of $\mathbf{h}_t$ (*Figure 3d*). We again find the hidden neuron activity to be low-dimensional, with approximate dimension $2.07 \pm 0.12$ (mean±s.e.), lending it to informative visualization using a PCA projection (Methods 5.4). Unlike the RNN, we observe the hidden neuron activity to be separated into several distinct clusters (*Figure 3e*). Exemplar input sequences cause $\mathbf{h}_t$ to rapidly transition between said clusters. Coloring the $\mathbf{h}_t$ by the sequence input at the present time step, we see the different inputs are what divide the hidden activity into distinct clusters, that we hence call *input-clusters* (*Figure 3f*). That is, the hidden neuron activity is largely dependent upon the most recent input to the network, rather than the accumulated evidence as we saw for the RNN. However, within each input-cluster, we also see a variation in $\mathbf{h}_t$ from accumulated evidence (*Figure 3e*). For the MPN (MPNpre), we now find only $0.21 \pm 0.01$ ($0.16 \pm 0.05$) of the hidden activity variance to be explained by accumulated evidence and $0.87 \pm 0.01$ ($0.80 \pm 0.03$) to be explained by the present input to the network (mean±s.e., Methods 5.4). (The MPNpre dynamics are largely the same of what we discuss here, see Sec. 5.4.2 for further discussion).

With the hidden neuron activity primarily dependent upon the current input to the network, one may wonder how the MPN ultimately outputs information dependent upon the entire sequence to solve the task. Like the other possible inputs to the network, the go signal has its own distinct input-cluster within which the hidden activities vary by accumulated evidence. Amongst all input-clusters, we find the readout vectors are highly aligned with the evidence variation within the go cluster (*Figure 3f*, inset). The readouts are then primed to distinguish accumulated evidence immediately following a go signal, as required by the task. (This idea leads to another intuitive visualization of the MPN behavior by asking what $\mathbf{h}_t$ would look like at any given time step if the most recent input is the go signal (*Figure 3—figure supplement 1e and f*)).

## MPNs encode accumulated evidence in the synapse modulations ($\mathbf{M}_t$)

Although the hidden neuron behavior is useful for comparison to the RNN and to understand what we might observe from neural recordings, information in the MPN is passed from one step to the next solely through the SM matrix (*Equation 2a*) and so it is also insightful to understand its dynamics. Flattening each $\mathbf{M}_t$ matrix, we can investigate the population dynamics in a manner identical to the hidden activity.

Once again, we find the variation of the SM matrix to be low-dimensional meaning we can visualize the evolution of its elements in its PCA space (*Figure 3g and h*). From exemplar sequences, we see that $\mathbf{M}_t$ appears to evolve in time along a particular direction as the input sequence is passed (*Figure 3g*). Perpendicular to this direction, we see a distinct separation of $\mathbf{M}_t$ values by accumulated evidence, very similar to what was seen for the RNN hidden activity (*Figure 3h*). Also like the RNN, the distinct evidence inputs tend to cause a change in the state in opposite directions (*Figure 3h*, inset). We also note the input that provides no evidence for either class and the go signal both cause sizable changes in the SM matrix.

Thus, since the state of the MPN is stored in the SM matrix, we see its behavior is much more similar to the dynamics of the hidden neuron activity of the RNN: each of the states tracks the accumulated evidence of the input sequence. Information about the relative evidence for each class is stored in the position of the state in its state space, and this information is continuously updated as new inputs are passed to the network by moving around said state space. Even so, the fact that the size of the MPN state seems to grow with time (a fact we confirm two paragraphs below) and has large deflections even for inputs that provide no evidence for either class make it stand apart from the dynamics of the RNN's state.

## The MPN and RNNs have distinct long-time behaviors

Given that $\mathbf{M}_t$ appears to get progressively larger as the sequence is read in (*Figure 3e*), one might wonder if there is a limit to its growth. More broadly, this brings up the question of what sort of

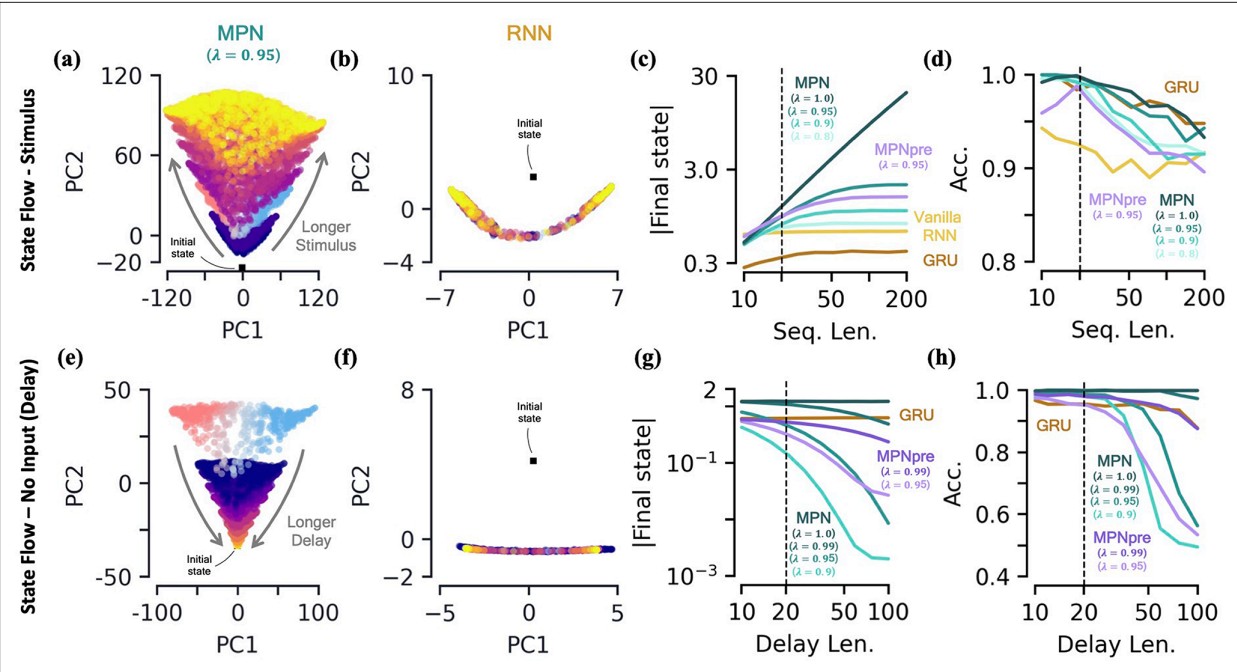

**Figure 4.** Two-class integration: long-time behavior of multi-plasticity network and RNNs. [a-d] *Flow of states under stimulus.* (a) The MPN state, $\mathbf{M}_t$, collected over 100 input sequences, colored by the stimulus length $t$, for $t = 10$ (dark blue) to 200 (yellow) time steps, with $\lambda = 0.95$. The $\mathbf{M}_t$ are projected onto their PCA space. The blue/red colored $\mathbf{M}_t$ are for $t = 20$ and are the same as those in *Figure 3f*. (b) Same as (a), for the Vanilla RNN states, $\mathbf{h}_t$, plotted in their PCA space. (c) For various sequences lengths, $T$, normalized magnitude of final states. Networks trained on a sequence length of $T = 20$ (dotted line). (d) Accuracy of the networks shown in (c) as a function of sequence length. [e-h] *Flow of states under zero input (delay input)*. (e) MPN states, $\mathbf{M}_t$, collected over 100 input sequences, colored by the delay length, 10 (dark blue) to 100 (yellow) time steps. The blue/red colored $\mathbf{M}_t$ are for zero delay. (f) Same as (e), for GRU states. (g) For various delay lengths, magnitude of final state. Networks trained with a delay of 20 (dotted line). (h) Accuracy of the networks shown in (i) as a function of delay length.

long-time behavior the MPN has, including any attractor structure. A full understanding of attractor dynamics is useful for characterizing dynamical systems. Attractors are often defined as the set of states toward which the network eventually flows asymptotically in time. As mentioned earlier, it is known that RNNs form low-dimensional, task-dependent attractor manifolds in their hidden activity space for integration-based tasks (*Maheswaranathan et al., 2019a*; *Aitken et al., 2020*). However, where the activity of a network flows to is dependent upon what input is being passed to the network at that time. For example, the network may flow to a different location under (1) additional stimulus input versus (2) no input. We will investigate these two specific flows for the MPN and compare them to RNNs.

We will be specifically interested in the flow of $\mathbf{M}_t$, the MPN's state, since from the previous section it is clear its dynamics are the closest analog of an RNN's state, $\mathbf{h}_t$. We train both the MPN and RNNs on a $T = 20$ integration task and then monitor the behavior of their states for stimuli lengths ranging from 10 to 200 steps. As might be expected from the dynamics in the previous section, we do indeed observe that $\mathbf{M}_t$'s components grow increasingly large with longer stimuli (*Figure 4a*). Meanwhile, the RNN's state appears to remain constrained to its line attractor even for the longest of input sequences (*Figure 4b*). To quantitatively confirm these observations, we look at the magnitude of the network's final state (normalized by number of components) as a function of the stimulus length (*Figure 4c*, Methods 5.5). As expected, since the RNNs operate close to an attractor, the final state magnitude does not increase considerably despite passing stimuli 10 times longer than what was observed in training. In contrast, the magnitude of the MPN state can change by several orders of magnitude, but its growth is highly dependent on the value of its $\lambda$ parameter that controls the rate of exponential decay (*Figure 4c*). (Although for any $\lambda < 1$ the modulations of the MPN will eventually stop growing, the ability of the SM matrix to grow unbounded is certainly biologically unrealistic. We introduced explicitly bounded modulations to see if this changed either the computational capabilities or behavior of the MPN, finding that both remained largely unchanged (Methods 5.4.5, *Figure 3—figure*

*supplement 2*)). As $\mathbf{M}_t$ grows in magnitude so does the size of its decay and eventually this decay will be large enough to cancel out the growth of $\mathbf{M}_t$ from additional stimuli input. For smaller $\lambda$ this decay is larger and thus occurs at shorter sequence lengths. Despite this saturation of state size, the accuracy of the MPN does not decrease significantly with longer sequence lengths (*Figure 4d*). These results also demonstrate the MPN (and RNNs) are capable of generalizing to both shorter and longer sequence lengths, despite being trained at a fixed sequence length.

## The MPN has a single, point-like attractor that its state uniformly decays toward

Another important behavior that is relevant to the operation of these networks is how they behave under no stimulus input. Such inputs occur if we add a delay period to the simple integration task, so we now turn to analyzing the MPN trained on a integration-delay task (*Figure 2c*). We again train MPNs with varying $\lambda$ and RNNs, this time on a task with a delay length of 20 time steps. (Vanilla RNNs trained on this task perform poorly due to vanishing gradients, so we omit them for this analysis. It is possible to train them by bootstrapping their training by gradually increasing the delay period of the task). During the delay period, the state of the MPNs decay over time (*Figure 4e*). Once again, since the RNNs operates close to its a line attractor manifolds, its state changes little over the delay period, other than flowing more towards the ends of the line (*Figure 4f*). Again, we quantify this behavior by monitoring the normalized final state magnitude as a function of the delay length (*Figure 4g*). We see the decay in the MPN's state is fastest for networks with smaller $\lambda$ and that the network with $\lambda = 1.0$ has no such decay.

Perhaps obvious in hindsight, the MPN's state will simply decay toward $\mathbf{M} = \mathbf{0}$ under no input. As such, for $\lambda < 1$, *the MPN has an attractor at $\mathbf{M} = \mathbf{0}$*, due to the exponential decay of its state built into its update expression. Since the evidence is stored in the magnitude of certain components of the SM matrix, a uniform decay across all elements maintains their relative size and does not decrease the MPN's accuracy for shorter delay lengths (*Figure 4h*). However, eventually the decay decreases the information stored in $\mathbf{M}_t$ enough that early inputs to the network will be indistinguishable from input noise, causing the accuracy of the network to plummet as well. The RNN, since it operates close to an attractor, has no appreciable decay in its final state over longer delays (*Figure 4g*). Still, even RNN attractors are subject to drift along the attractor manifold and we do eventually see a dip in accuracy as well (*Figure 4h*).

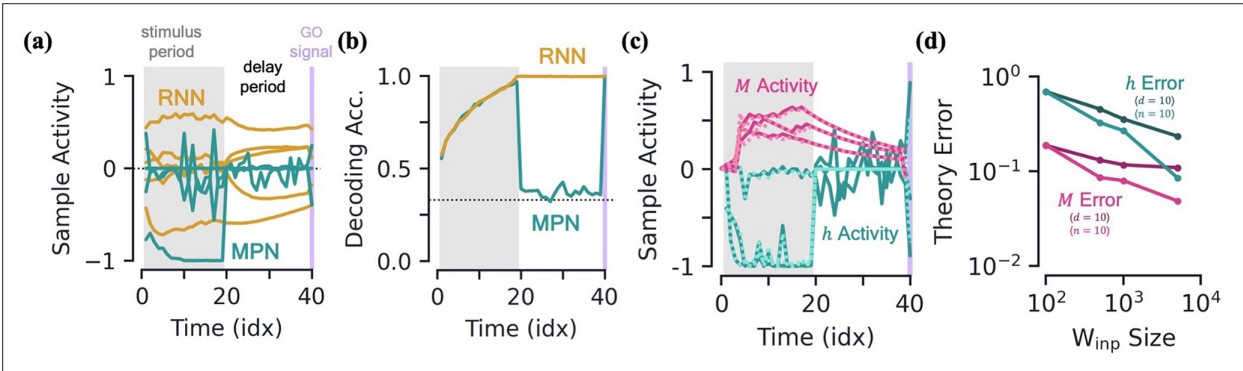

**Figure 5.** Integration with delay and analytical predictions. [**a-b**] *MPN and RNN hidden activity behavior during a integration-delay task*. The left shaded grey region represents the stimulus period; followed by the delay period with a white background; and finally the thin purple-shaded region representing the response to the go signal. (**a**) Sample hidden neuron activity for the MPN and RNN. (**b**) Decoding accuracy on the hidden neuron activity of the MPN and RNN as a function of time (Methods 5.5). Dotted line represents chance accuracy. [**c-d**] *Analytical MPN approximations.* (**c**) Exemplar hidden neuron (teal) and SM matrix (pink) activity as a function of time (solid lines) and their analytical predictions (dotted lines) from *Equations 4 and 5*. Shading is the same as (**a**) and (**b**). (**d**) Overall accuracy of of theoretical predictions as a function of the size of $\mathbf{W}_{\mathrm{inp}}$ with either $n$ or $d$ fixed (Methods 5.5).

The online version of this article includes the following figure supplement(s) for figure 5:

**Figure supplement 1.** More on delay dynamics of the MPN and RNN.

## MPNs are 'activity silent' during a delay period

We have seen the MPN's state decays during a delay period, here we investigate what we would observe in its hidden neurons during said period. Since the MPN's hidden activity primarily depends on the present input, during the delay period when no input is passed to the network (other than noise), we expect the activity to be significantly smaller (*Sugase-Miyamoto et al., 2008*). Indeed, at the start of the delay period, we see a few of the MPN's hidden neuron activities quickly drops in magnitude, before spiking back up after receiving the go signal (*Figure 5a*). Meanwhile, at the onset of the delay, the RNN's hidden layer neurons quickly approach finite asymptotic values that remain fairly persistent throughout the delay period (although see *Orhan and Ma, 2019* for factors that can affect this result). The aforementioned behaviors are also seen by taking the average activity magnitude across the entire population of hidden neurons in each of these networks (*Figure 5—figure supplement 1a*). Reduced activity during delay periods has been observed in working memory experiments and models and is sometimes referred to as an 'activity-silent' storage of information (*Barak et al., 2010*; *Stokes, 2015*; *Lundqvist et al., 2018*). It contrasts with the 'persistent activity' exhibited by RNNs that has been argued to be more metabolically expensive (*Attwell and Laughlin, 2001*).

To further quantify the degree of variation in the output information stored in the hidden neuron activity of each of these networks as a function of $t$, we train a decoder on the $\mathbf{h}_t$ (Methods 5.5; *Masse et al., 2019*). Confirming that the MPN has significant variability during its stimulus and delay periods, we see the MPN's decoding accuracy drops to almost chance levels at the onset of the delay period before jumping back to near-perfect accuracy after the go signal (*Figure 5b*). Since the part of the hidden activity that tracks accumulated evidence is small, during this time period said activity is washed out by noise, leading to a decoding accuracy at chance levels. Meanwhile, the RNN's hidden neuron activity leads to a steady decoding accuracy throughout the entire delay period, since the RNN's state just snaps to the nearby attractor, the position along which encodes the accumulated evidence. Additionally, the RNN's trained decoders maintain high accuracy when used at different sequence times, whereas the cross-time accuracy of the MPN fluctuates significantly more (*Figure 5—figure supplement 1e, f and g*). The increased time-specificity of activity in the MPN has been observed in working memory experiments (*Stokes et al., 2013*). (Additional measures of neuron variability and activity silence are shown in the supplement (*Figure 5—figure supplement 1*)).

## Analytical confirmation of MPN dynamics

It is possible to analytically approximate the behavior of the MPN's $\mathbf{h}_t$ and $\mathbf{M}_t$ at a given time step. Details of the derivation of these approximations is given in Methods 5.3. Briefly, the approximation relies on neglecting quantities that are made small with an increasing number of neurons in either the input or hidden layer. The net effect of this is that *synaptic modulations are small* and thus can be neglected at leading-order approximations. Explicitly, the approximations are given by

$$\mathbf{h}_t \approx \tanh\left(\underbrace{\mathbf{W}_{\text{inp}}\mathbf{x}_t}_{\text{leading}} + \underbrace{\sum_{s=1}^{t-1} \eta\lambda^{s-1}\tanh\left(\mathbf{W}_{\text{inp}}\mathbf{x}_{t-s}\right)\odot\left(\mathbf{W}_{\text{inp}}\left(\mathbf{x}_{t-s}\odot\mathbf{x}_t\right)\right)}_{\text{sub-leading}}\right), \tag{4}$$

$$\mathbf{M}_t \approx \lambda\mathbf{M}_{t-1} + \eta\tanh\left(\mathbf{W}_{\text{inp}}\mathbf{x}_t\right)\mathbf{x}_t^T, \tag{5}$$

where, in the first expression, we have indicated the terms that are the leading and sub-leading contributions. These approximation do quite well in predicting the element-wise evolution of $\mathbf{h}_t$ and $\mathbf{M}_t$, although are notably bad at predicting the hidden activity during a delay period where it is driven by only noise (*Figure 5c*). We quantify how good the approximations do across the entire test set and see that they improve with increasing input and hidden layer size (*Figure 5d*).

These simplified analytical expressions allow us to understand features we've qualitatively observed in the dynamics of the MPN. Starting with the expression for $\mathbf{h}_t$, we see the leading-order contributions comes from the term $\mathbf{W}_{\text{inp}}\mathbf{x}_t$, which is solely dependent upon the current input, i.e. *not* the sequence history. Comparing to the exact expression, *Equation 2a*, the leading-order approximation is equivalent to taking $\mathbf{M}_{t-1} = \mathbf{0}$, so *at leading-order the MPN just behaves like a feedforward network*. This explains why we see the input-dependent clustering in the $\mathbf{h}_t$ dynamics: a feedforward network's activity is only dependent on its current input. Meanwhile, the sub-leading term depends

on all previous inputs $(\mathbf{x}_{t-1}, \mathbf{x}_{t-2}, \ldots, \mathbf{x}_1)$, which is why the individual input-clusters vary slightly by accumulated evidence.

From the approximation for $\mathbf{M}_t$, we see its update from one time step is solely dependent upon the current input as well. Without the $\lambda$ term, $\mathbf{M}_t$ simply acts as an accumulator that counts the number of times a given input has been passed to the network – exactly what is needed in an integration-based task. In practice, with $\lambda < 1$, the contribution of the earlier inputs of the network to $\mathbf{M}_t$ will slowly decay.

The MPNpre modulation updates are significantly simpler to analyze since they do not depend on the hidden activity, and thus the recurrent dependence of $\mathbf{M}_t$ on previous inputs is much more straightforward to evaluate. The equivalent expressions of *Equations 4 and 5* for the MPNpre are given in Sec. 5.3.

## Capacity, robustness, and flexibility

Having established an understanding of how the dynamics of the MPN compares to RNNs when trained on a simple integration-delay task, we now investigate how their different operating mechanisms affect their performance in various settings relevant to neuroscience.

### MPNs have comparable integration capacity to GRUs and outperform Vanilla RNNs

Given their distinct state storage systems, it is unclear how the capacity of the MPN compares to RNNs on integration-based tasks. In RNNs, the capacity to store information has been linked to the number of synapses/parameters and the size of their state space (*Collins et al., 2016*). For example, since we know that RNNs tend to use an approximate $(N-1)$-dimensional attractor manifold in their hidden activity space to solve an N-class task (*Aitken et al., 2020*), one might expect to see a drop in accuracy for $N-1 > n$, with $n$ the number of hidden neurons. To investigate state storage capacity in the MPN and RNNs, we limit the number of adjustable parameters/synaptic connections by making the number of neurons in each layer small, specifically taking the number of input and hidden neurons to be $d = 10$ and $n = 3$, respectively.

We observe the MPN and GRU (and MPNpre) are capable of training to accuracies well above chance, even for $N \geq 3$ classes, while the Vanilla RNN's accuracy quickly plummets beyond $N = 2$ (*Figure 6a*, *Figure 6—figure supplement 1a*). The size of the MPN state scales as $nd$, and indeed we see the the accuaracies receive a small bump, becoming more comparable to that of the GRU, when the input dimension is increased from $d = 10$ to 40. (For $n = 3$ and fixed $d$, the number of trainable parameters in the MPN is smaller than that of the RNNs, see *Table 1*. With $d = 40$, the MPN has a number of trainable parameters more comparable to that of the $d = 10$ GRU (125 and 126, respectively)).

A second way we can test for integration information capacity is to increase the length of the input sequences in the task. Across the board, both the MPN, MPNpre, and RNNs are capable of learning input sequences up to length $T = 200$ at high accuracy (*Figure 6b*, *Figure 6—figure supplement 1b*). Although differences are of only a few percent, the MPN is capable of learning to integrate relatively long sequences at a level greater than Vanilla RNNs, but not quite as good as GRUs.

### MPNs can operate at high input noise and minimal training

To test the robustness of the MPN's dynamics we make the integration task harder in two different ways. First, we add increasingly more noise to the inputs to the network. Even for networks trained with a relatively small amount of noise, we find both the MPN and RNNs are capable of achieving near-perfect accuracy on the two-class task up to when noise is a comparable magnitude to the signal (*Figure 6c*). Continuing to increase the size of the noise, we see all networks eventually fall to chance accuracy, as expected. Notably, the RNNs maintain higher accuracy for a slightly smaller ratio of signal to noise magnitude. This might be expected given the RNN's operation close to attractor manifolds, which are known to be robust to perturbations such as noisy inputs (*Vyas et al., 2020*).

Second, we aim to understand if the MPN's intrinsic dynamics at initialization allow it to perform integration with a minimal adjustment of weights. We test this by freezing all internal parameters at their initialization values and only training the MPN's readout layer. It is well known that RNNs with a large number of hidden layer neurons have varied enough dynamics at initialization that simply

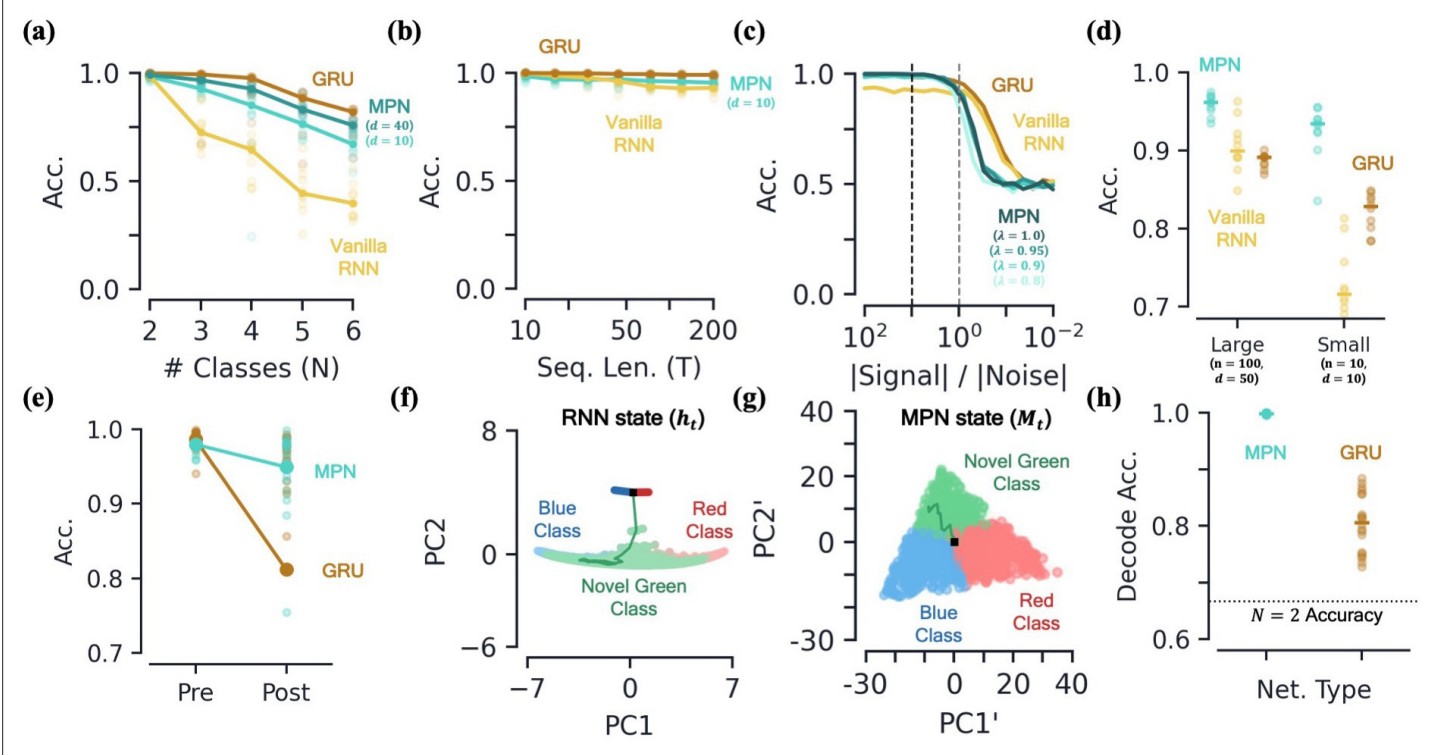

**Figure 6.** Capacity, robustness, and flexibility of the MPN. [a-d] Accuracy of the MPN and RNNs as a function of several measures that make the integration task more difficult, see *Figure 6—figure supplement 1 for MPNpre results*. (a) For very small networks ($d = 10$, $n = 3$), number of classes in integration task ($N$) with fixed sequence length, $T = 20$ (Methods 5.5). (b) Also for very small networks, length of integration task ($T$) with fixed number of classes, $N = 2$. (c) Ratio of signal and noise magnitudes of the input (Methods 5.5). Networks were trained at a ratio of 10 (dotted black line), the dotted grey line represents a ratio of 1.0. (d) Networks trained with all parameters frozen at initialization values except for readout matrix (and $\eta = 1.0$, $\lambda = 0.95$). [e-f] *Flexibility to learn new tasks*. (e) Accuracy on a two-class integration-delay task pre- and post-training on a novel two-class integration-delay task. Thick lines/dots show averages, raw trial data is scattered behind. (f) Hidden activity, colored by sequence label, for a GRU trained on a two-class integration task (red/blue classes) when a novel class (green) is introduced *without* additional training. Activity collected over 1000 input sequences, plotted in their PCA space. Also shown are readouts, initial state, and sample trajectory. (g) Same as (f), for the MPN state, $\mathbf{M}_t$(h) Decoding accuracy of the final states of the MPN and GRU when a novel class introduced, again without training. Dotted line represents accuracy that would be achieved for perfect classification of only the 2 familiar classes.

The online version of this article includes the following figure supplement(s) for figure 6:

**Figure supplement 1.** Additional performance comparisons, including additive MPN.

adjusting the readout layer allows them to accomplish a wide variety of tasks (*Maass et al., 2002*; *Jaeger and Haas, 2004*; *Lukoševičius and Jaeger, 2009*). Such settings are of interest to the neuroscience community since the varied underlying dynamics in random networks allow for wide variety of responses, matching the observed versatility of certain areas of the brain, for example the neocortex (*Maass et al., 2002*). Since the $\eta$ and $\lambda$ parameters play especially important roles in the MPN, we fix them at modest values, namely $\eta = 1.0$ and $\lambda = 0.95$. (We do not see a significant difference in accuracy for $\eta = -1.0$, that is an anti-Hebbian update rule). For two different layer sizes, we find all networks are capable of training in this setup, but the MPN consistently outperforms both RNNs (*Figure 6d*). Notably, even the MPN with significantly less neurons across its input/hidden layers outperforms the RNNs. These results suggest that the intrinsic computational structure built into the MPN from its update expressions allow for it to be particularly good at integration tasks, even with randomized synaptic connections.

## MPNs are flexible to taking in new information

The flexibility of a network to learn several tasks at once is an important feature in both natural and artificial neural networks (*French, 1999*; *McCloskey and Cohen, 1989*; *McClelland et al., 1995*; *Kumaran et al., 2016*; *Ratcliff, 1990*). It is well-known that artificial neural networks can suffer from large drops

**Table 1.** Parameter and operation counts for various networks.

The number of neurons in the input, hidden, and output layers are $d$, $n$, and $N$, respectively. Note these counts do not include parameters of the readout layer, since said layer contributes the same number of parameters for each network ($N(n + 1)$ and $Nn$ with and without a bias) and are for fixed initial states.

| Network | Trainable Parameters | State update operations |
|---|---|---|
| 2-layer fully connected | $n(d+1)$ | $\mathcal{O}\left(nd^2\right)$ |
| MPN | $n(d+1)+2$ | $\mathcal{O}\left(nd^2\right)$ |
| MPNpre | $n(d+1)+2$ | $\mathcal{O}\left(nd^2\right)$ |
| Vanilla RNN | $n(n+d+1)$ | $\mathcal{O}\left(n^3\right)+\mathcal{O}\left(nd^2\right)$ |
| GRU | $3n(n+d+1)$ | $\mathcal{O}\left(3n^3\right)+\mathcal{O}\left(3nd^2\right)$ |
| LSTM | $4n(n+d+1)$ | $\mathcal{O}\left(4n^3\right)+\mathcal{O}\left(4nd^2\right)$ |

in accuracy when learning tasks sequentially. This effect has been termed *catastrophic forgetting*. For example, although it is known artificial RNNs are capable of learning many neuroscience-related tasks at once, this is not possible without interleaving the training of said tasks or modifying the training and/or network with continual-learning techniques (*Yang et al., 2019*; *Duncker et al., 2020*). Given the minimal training needed for an MPN to learn to integrate, as well as its task-independent attractor structure, here we test if said flexibility also extends to a sequential learning setting.

To test this, we train the MPN and GRU on a two-class integration-delay task until a certain accuracy threshold is met and then train on a *different* two-class integration-delay task until the accuracy threshold is met on the novel data. Afterwards, we see how much the accuracy of each network on the original two-class task falls. (Significant work has been done to preserve networks from such pitfalls by, for example, modifying the training order (*Robins, 1995*) or weight updates (*Kirkpatrick et al., 2017*). Here, we do not implement any such methods, we are simply interesting in how the different operating mechanisms cause the networks to behave 'out of the box'). We find that the MPN loses significantly less accuracy than the GRU when trained on the new task (*Figure 6e*). Intuitively, an RNN might be able to use the same integration manifold for both integration tasks, since they each require the same capacity for the storage of information. The state space dimensionality of the MPN and GRU do not change significantly pre- and post-training on the novel data (Methods 5.5). However, we find the line attractors reorient in state space before and after training on the second task, on average shifting by $23 \pm 3$ degrees (mean±s.e.). Since the MPN has a task-agnostic attractor structure, it does not change in the presence of new data.

To understand the difference of how these two networks adapt to new information in more detail, we investigate how the MPN and RNN dynamics treat a novel input, for example how networks trained on a two-class task behave when suddenly introduced to a novel class. For the RNN, the novel inputs to the network do not cause the state to deviate far from the attractor (*Figure 6f*). The attractors that make the RNNs so robust to noise are their shortcoming when it comes to processing new information, since anything it hasn't seen before is simply snapped into the attractor space. Meanwhile for the MPN, the minimal attractor structure means new inputs have no problem deflecting the SM matrix in a distinct direction from previous inputs (*Figure 6g*). To quantify the observed separability of the novel class we train a decoder to determine the information about the output contained in the final state of each network (still with no training on the novel class). The MPN's states have near-perfect separability for the novel class even before training, $0.998 \pm 0.001$ accuracy, while the GRU has more trouble separating the new information, $0.812 \pm 0.009$ accuracy (mean±s.e., *Figure 6h*). Hence, out of the box, the MPN is primed to take in new information.

## Additional tasks

Given the simplicity of the MPN, it may be called into question if such a setup is capable of learning anything beyond the simple integration-delay tasks we have presented thus far. Additionally, if it is capable of learning other tasks, how its dynamics may change in such settings is also of interest. To address these questions, in this section we train and analyze the MPNs on additional integration tasks studied in neuroscience, many of which require the network to learn more nuanced behavior such as context. Additionally, dynamics of networks trained on prospective and true-anti contextual tasks are shown in *Figure 7—figure supplement 1* (Methods 5.1).

### Retrospective contextual integration

Integration with context is a well-known task in the neuroscience literature (*Mante et al., 2013*; *Panichello and Buschman, 2021*). In this setup, two independent two-class integration-delay *subtasks* are passed to the network at once (inputs are concatenated) and the network must provide the correct label to only one of the subtasks based on the contextual input it receives (i.e. report subtask 1 or 2). Here, we discussing the retrospective case, where the context comes *after* the stimuli of the subtasks, in the middle of the delay period (*Figure 7a*).

We find the MPN is easily capable of learning this task and again achieves near-perfect accuracy, on average 98.4%. To understand how the MPN is capable of processing context, we can investigate how its dynamics change from that of the simple integration task analyzed previously. Once again turning to the state of MPN to see what information it encodes, during the time-period where the subtask stimuli are being input we see $\mathbf{M}_t$ holds information about both the integration subtasks simultaneously (*Figure 7b*). That is, we see the same continuum that encodes the relative evidence between the two classes that we saw in the single task case earlier (*Figure 3f*), but for both of the integration subtasks. Furthermore, these two one-dimensional continua lie in distinct subspaces, allowing a single location in the two-dimensional space to encode the relative evidences of both subtasks at once. This make sense from the perspective that the network does not yet know which information is relevant to the output, so must encode information from each subtask prior to seeing the context.

When the context is finally passed to the network, we see its state space becomes increasingly separated into two clusters that correspond to whether the context is asking to report the label from subtask 1 or 2 (*Figure 7c*). Separating the states into distinct regions of state space allows for them to be processed differently when converted to hidden and output activity, and we now show this is how the MPN solves the task. We can quantify the difference the separation induced by contextual input produces by looking at the how each states gets converted into an output and how this changes with context. We define a subtask's *readout difference* such that more positive values means the two classes belonging to the subtask are easier to distinguish from one another via the readout vectors, that is what is needed to solve the subtask (Methods 5.5). Prior to the context being passed, we see the readout difference increases for both subtasks as evidence is accumulated (*Figure 7d*). As soon as the contextual input is passed, the readout difference for the subtask that is now irrelevant (the one that doesn't match the context, dotted line) immediately plummets, while that of the relevant subtask (the one that matches the context, solid line) increases (*Figure 7d*). (An alternative way to quantify this difference is to compare each subtask's direction of evidence variation (*Figure 7b*) in hidden activity to that of the readout directions. We find that, after context is passed to the network, the now-irrelevant subtask's evidence variation becomes close to perpendicular to the readouts, meaning their difference is almost irrelevant to the output neurons (*Figure 7—figure supplement 1d and e*)). After a final delay period, the go signal is passed to the network and these two separate clusters of state space are readout distinctly, allowing the network to output the label of the appropriate subtask.

### Continuous integration

Thus far, we have investigated integration tasks where evidence comes in discrete chunks, but often evidence from stimuli can take on continuous values (*Mante et al., 2013*). In this task, the network receives continuous inputs and must determine if the input was drawn from a distribution with positive or negative mean (*Figure 7e*). Evidence is again passed to the network through a random binary vector, but the vector is multiplied by the continuous signal. The MPN is once again able to achieve near perfect accuracy on this task.

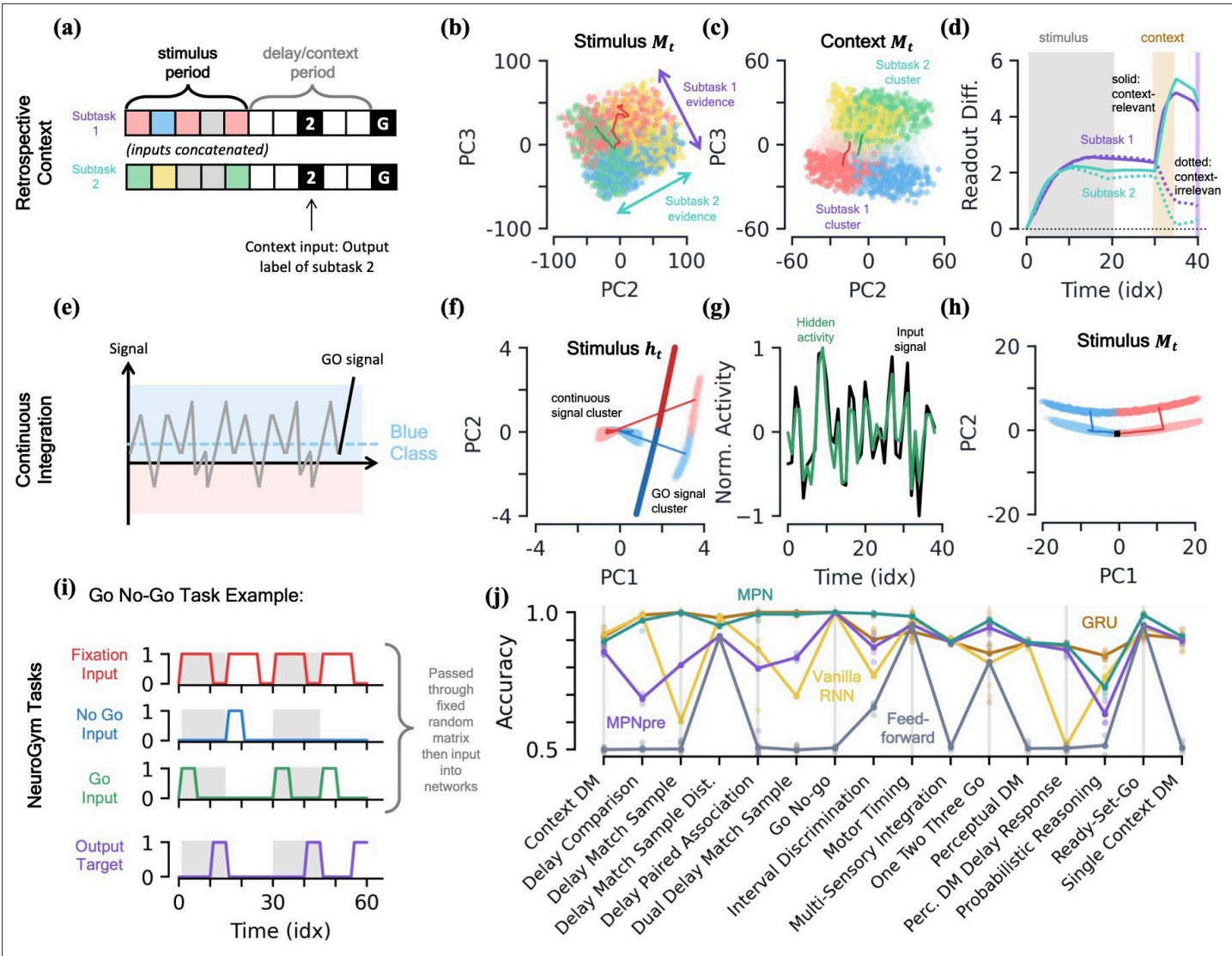

**Figure 7.** MPN dynamics on additional tasks. [a-d] *Retrospective context task.* (**a**) Schematic of example sequence. (**b**) MPN states during the initial stimulus period, projected into their (flattened) PCA space, colored by label and subtask (subtask 1: red/blue, subtask 2: green/yellow). Two example trajectories are shown. (**c**) Same as (**b**), for states during the time period where the network is passed context (5 time steps). States at end of time period colored darker for clarity. (**d**) The readout difference of the MPN as a function of sequence time (Methods 5.5). More positive values correspond to the MPN better distinguishing the two classes for the subtask. Solid lines correspond to when the subtask is chosen by the context, dotted lines to when the subtask is not chosen by the context. Grey, yellow, and white background correspond to stimulus, context, and delay periods, respectively. [e-h] *Continuous integration task.* (**e**) Schematic of example sequence. (**f**) Hidden neuron activity, colored by sequence label, projected into their PCA space. Thick lines are readout vectors of corresponding class. (**g**) Normalized input (black) and example hidden neuron (green) activities, as a function of time (Methods 5.5). (**h**) MPN states, colored by sequence label, projected into their PCA space. Two example trajectories are shown, black square is the initial state. [i-j] *NeuroGym tasks* (**French, 1999**). (**i**) The beginning of an example sequence of the Go No-Go task. The grey/white shading behind the sequence represents distinct stimulus/delay/decision periods within the example sequence. (**j**) Performance of the various networks we investigate across 19 supervised learning NeuroGym tasks (Methods 5.5).

The online version of this article includes the following figure supplement(s) for figure 7:

**Figure supplement 1.** More integration tasks, supplemental figures.

Since all evidence is scalar multiples of a random input vector, the hidden neuron activity prior to the go signal now exists in a single cluster as opposed to the distinct input-clusters we saw for the discrete case (*Figure 7f*, *Figure 7—figure supplement 1j*). The go signal again has its own separate cluster, within which the hidden state varies with the total evidence of the sequence and is well-aligned with the readout vectors (*Figure 7f*). Although the dynamics of this task may look more similar

to the line-attractors, we again note that the hidden neuron activity largely tracks input activity rather than accumulated evidence, unlike an RNN (*Figure 7g*). The accumulated evidence is still stored in the SM matrix, $\mathbf{M}_t$. In the low-dimensional space, the state moves along a line to track the relative evidence between the two classes, before jumping to a separate cluster when the go signal is passed (*Figure 7h*). Again note this is distinct from line attractor dynamics of the RNN, since in the absence of stimulus during a delay period, $\mathbf{M}_t$ will still exponentially decay back towards its baseline value at $\mathbf{M} = \mathbf{0}$.

## NeuroGym tasks

NeuroGym is a collection of tasks with a common interface that allows for rapid training and assessment of networks across many neuroscience-relevant computations (*Molano-Mazon et al., 2022*). We train the MPN, MPNpre, and RNNs on 19 different supervised learning NeuroGym tasks. Whereas the tasks we have considered thus far only require the networks output task-relevant information after a go signal, the NeuroGym tasks require the network to give a specific output at all sequence time steps (*Figure 7i*). Since several of the NeuroGym tasks we investigate have very low-dimensional input information (the majority with $d \leq 3$), we pass pass all inputs through a fixed random matrix and added bias before passing them into the relevant networks (Methods 5.1). Across all 19 tasks we investigate, we find that the MPN performs at levels comparable to the GRU and VanillaRNN (*Figure 7j*). Despite its simplified modulation mechanism, the MPNpre also achieves comparable performance on the majority of tasks. Since we compare networks with the same number of input, hidden, and output neurons, we again note that the MPN and MPNpre have significantly fewer parameters than their RNN counterparts (see *Table 1*).

## Discussion

In this work, we have thoroughly explored the trained integration dynamics of the MPN, a network with multiple forms of plasticity. It has connections between neurons that are effectively a product between two terms: (1) the $\mathbf{W}_{\text{inp}}$ matrix, trained in a supervised manner with backpropagation and assumed to be constant during input sequences and (2) the synaptic modulation matrix, $\mathbf{M}$, which has faster dynamics and evolves in an unsupervised manner. We analyzed MPNs without recurrent connections so that they have to rely solely on synaptic dynamics for the storage and updating of short-timescale information. Unlike an RNN, we have found the dynamics of the hidden neurons in the MPN primarily track the present input to the network, and only at subleading-order do we see them encode accumulated evidence. This makes sense from the point of view that the hidden neurons have two roles in the MPN: (1) they connect directly to the readouts and must hold information about the entire sequence for the eventual output, but also (2) they play a role in encoding the input information to update the $\mathbf{M}_t$ matrix. We also investigated the MPNpre, a network whose modulations depend only on presynaptic activity. Despite this simplified update mechanism, we find the MPNpre is often able to perform computations at the level of the MPN and operates using qualitatively similar dynamics.

The synaptic modulations, contained in the SM matrix, $\mathbf{M}_t$, encode the accumulated evidence of input sequence through time. Hence, the synaptic modulations play the role of the state of the MPN, similar to the role the hidden activity, $\mathbf{h}_t$, plays in the RNN. Additionally, we find the MPN's state space has a fundamentally different attractor structure than that of the RNN: the uniform exponential decay in $\mathbf{M}_t$ imbues the state space with a single point-like attractor at $\mathbf{M} = \mathbf{0}$. Said attractor structure is task-independent, which significantly contrasts the manifold-like, task-dependent attractors of RNNs. Despite its simplistic attractor structure, the MPN can still hold accumulated evidence over time since its state decays slowly and uniformly, maintaining the relative encoding of information (*Figure 4*).

Although they have relatively simple state dynamics, across many neuroscience-relevant tests, we found the MPN and MPNpre were capable of performing at comparable levels to the VanillaRNN and GRU, sometimes even outperforming their recurrent counterparts. The exception to this was noise robustness, where the RNNs' attractor structure allows them to outperform the MPN in a relatively small window of signal to noise ratio. However, the MPN's integration dynamics that rely on its minimal attractor structure allowed it to outperform RNNs in both minimal training and sequential-learning settings. Altogether, we find such performance surprising given the simplicity of the MPN and MPNpre. Unlike the highly designed architecture of the GRU, these modulation-based networks

operate using relatively simple biological mechanisms, with no more architectural design than the simplest feedforward neural networks.

While this study focuses on theoretical models with a general synapse-specific change of strength on shorter timescales than the structural changes implemented via backpropagation, we can hypothesize potential mechanistic implementations for the MPN and MPNpre. The MPNpre's presynaptic activity dependent modulations can be mapped to STSP, which is largely dependent on availability and probability of release of transmitter vesicles (*Tsodyks and Markram, 1997*). As an example, the associative modulations and backpropagation present in MPN could model the distinct plasticity mechanisms underlying early and late phases of LTP/LTD (*Baltaci et al., 2019*; *Becker and Tetzlaff, 2021*). The early phase of the LTP can be induced by rapid and transient Calmodulin-dependent Kinase II (CaMKII) activation, sometimes occurring within $\sim 10$ s (*Herring and Nicoll, 2016*) and which can last for $< 1$ min (*Lee et al., 2009*). CaMKII is necessary and sufficient for early LTP induction (*Silva et al., 1992*; *Pettit et al., 1994*; *Lledo et al., 1995*) and is known to be activated by Ca2$^+$ influx via N-methyl-D-aspartate receptors (NMDAR), which open primarily when the pre- and postsynaptic neurons activate in quick succession (*Herring and Nicoll, 2016*). This results in insertions of AMPA receptors in the synapses within 2 min resulting in changes in postsynaptic currents (*Patterson et al., 2010*). These fast transitions that occur on scales of minutes can stabilize and have been observed to decay back to baseline within a few hours (*Becker and Tetzlaff, 2021*). Altogether, these associative modulations leading to AMPA receptor exocytosis that can be sensitive to stimulus changes on the order of minutes can be represented by the within-sequence changes in modulations of the MPN. The AMPA receptor decay back to baseline corresponds to a decay of the modulations (via the $\lambda$ term) corresponding to hours. Subsequently, based on more complex mechanisms that involve new protein expression and can affect the structure, late phase LTP can stabilize changes over much longer timescales (*Baltaci et al., 2019*). If these mechanisms bring in additional factors that can be used in credit assignment across the same synapses affected by early phase LTP, they would map to the slow synaptic changes represented by backpropagation in the MPN's weight adjustment (*Lillicrap et al., 2020*).

The simplicity of the MPN and MPNpre leaves plenty of room for architectural modifications to either better match onto biology or improve performance. Foremost among such modifications is to combine recurrence and dynamic synapses into a single network. The MPN/MPNpre and Vanilla RNN are subnetworks of this architecture, and thus we already know this network could exhibit either of their dynamics or some hybrid of the two. In particular, if training is incentivized to find a solution with sparse connections or minimal activity, the MPN that computes with no recurrent connections and activity silence could be the preferred solution. The generality of the MPN/MPNpre's synaptic dynamics easily allows for the addition of such dynamic weights to any ANN layer, including recurrent layers (*Orhan and Ma, 2019*; *Ballintyn et al., 2019*; *Burnham et al., 2021*). Finally, adding synaptic dynamics that vary with neuron or individual synapses would also be straightforward: the scalar $\lambda$ and $\eta$ parameters that are uniform across all neurons can be replaced by a vector or matrix equivalents that can be unique for each pre- or postsynaptic neuron or synapse (*Rodriguez et al., 2022*). A non-uniform decay from vector/matrix-like $\lambda$ would allow the MPN have a more nuanced attractor structure in its state space. Indeed, the fact that the SM matrix decays to zero resembles the fading memory of reservoir computing models and there it has been shown multiple timescales facilitate computation (*de Sá et al., 2007*).

## Methods

As in the main text, throughout this section we take the number of neurons in the input, hidden, and output layers to be be $d$, $n$, and $N$, respectively. We continue to use uppercase bold letters for matrices and lowercase bold letters for vectors. We use $i, j = 1, \ldots, n$ to index the hidden neurons and $I, J = 1, \ldots, d$ to index the input neurons. For components of matrices and vectors, we use the same non-bolded letter, e.g. $M_{iJ,t}$ for $\mathbf{M}_t$ or $x_{I,t}$ for $\mathbf{x}_t$.

### Supporting code

Code for this work can be found at: https://github.com/kaitken17/mpn (copy archived at *Aitken and Mihalas, 2023*). We include a demonstration of how to implement a *multi-plasticity layer*, which allows

one to incorporate the synaptic modulations used in this work into any fully-connected ANN layer. This allows one to easily generalize the MPN to, say, deeper networks or networks with multi-plastic recurrent connections.

## Tasks

### Simple integration task

The simple integration task is used throughout this work to establish a baseline for how MPNs and RNNs learn to perform integration. It is inspired by previous work on how RNNs learn to perform integration in natural language processing tasks, where it has been shown they generate attractor manifolds of a particular shape and dimensionality (*Maheswaranathan et al., 2019a*; *Maheswaranathan and Sussillo, 2020*; *Aitken et al., 2020*).

The N-class integration task requires the network to integration evidence from multiple classes over time and determine the class with the most evidence (*Figure 2a*). Each example from the task consists of a sequence of $T$ input vectors, $\mathbf{x}_1, \ldots, \mathbf{x}_T$, passed to the network one after another. We draw possible inputs at a given time step from a bank of *stimulus inputs*, $\{\text{evid}_1, \text{evid}_2, \ldots, \text{evid}_N, \text{null}\}$. Here, 'evid$_m$' corresponds to one unit of evidence for the mth class. The 'null' input provides evidence for none of the classes. Each sequence ends in an 'go signal' input, letting the network know an output is expected at that time step. All the stimulus inputs have a one-to-one mapping to a distinct random binary vector that has an expected magnitude of 1 (*Figure 2b*, see below for details). Finally, each example has an integer label from the set $\{1, \ldots, N\}$, corresponding to the class with the most evidence. The network has correctly learned the task if its largest output component at time $T$ is the one that matches each example's label.

The input sequence examples are randomly generated as follows. For a given sequence of an N-class task, the amount evidence for each class in the entire sequence can be represented as an N-dimensional *evidence vector*. The mth element of this vector is the number of evid$_m$ in the given sequence. For example, the three-class sequence of length $T = 6$, 'evid$_2$, evid$_1$, null, evid$_1$, evid$_3$, go' has an evidence vector $(2, 1, 1)$. The sequences are randomly generated by drawing them from a uniform distribution *over possible evidence vectors*. That is, for a given $N$ and $T$, we enumerate all possible evidence vectors and draw uniformly over said set. Sequences that have two or more classes tied for the most evidence are eliminated. Then, for a given evidence vector, we draw uniformly over sequences that could have generated said vector (Note that simply drawing uniformly over the bank of possible inputs significantly biases the inputs away from sequence with more extreme evidence differences. This method is still biased toward evidence vectors with small relative differences, but much less so than the aforementioned method). Unless otherwise stated, we generally consider the case of $T = 20$ throughout this work.

To generate the random binary vectors that map to each possible stimulus input, each of their elements is independently drawn uniformly from the set $\{0, \sqrt{2/d}\}$. The binary vectors are normalized by $\sqrt{2/d}$ so they have an expected magnitude of 1,

$$\mathbb{E}\|\mathbf{x}\|_2^2 = \sum_{I=1}^{d} \mathbb{E}\, x_I^2 = \sum_{I=1}^{d} \frac{1}{2}\left(\sqrt{\frac{2}{d}}\right)^2 = 1\,. \tag{6}$$

where $\|\cdot\|_2$ denotes L2-normalization. Note the expected dot product between two such vectors is

$$\mathbb{E}\left(\mathbf{x} \cdot \mathbf{x}'\right) = \sum_{I=1}^{d} \mathbb{E}\left(x_I x_I'\right) = \sum_{I=1}^{d} \frac{1}{4}\left(\sqrt{\frac{2}{d}}\right)^2 = \frac{1}{2}\,. \tag{7}$$

Often, we will take the element-wise (Hadamard) product between two input vectors, the expected magnitude of the resulting vector is

$$\mathbb{E}\|\mathbf{x} \odot \mathbf{x}'\|_2^2 = \sum_{I=1}^{d} \mathbb{E}\left(x_I x_I'\right)^2 = \sum_{I=1}^{d} \frac{1}{4}\left(\sqrt{\frac{2}{d}}\right)^4 = \frac{1}{d}\,, \tag{8}$$

where we have used the fact that the only nonzero element of the element-wise product occurs when the elements are both $\sqrt{2/d}$. The fact that this product scales as $1/d$ will be useful for analytical approximations later on.

Notably, since this task only requires a network to determine the class with the *most* evidence (rather than the absolute amount of evidence for each class), we claim this task can be solved by keeping track of $N-1$ relative evidence values. For example, in a two-class integration task, at a minimum the network needs to keep track of a single number representing the relative evidence between the two classes. For a three-class task, the network could keep track of the relative evidence between the first and second, as well as the second and third (from which, the relative evidence between the first and third could be determined). This generalizes to $N-1$ for an N-class integration class.(The range of numbers the network needs to keep track of also scales with the length of the sequence, $T$. For instance, in a two-class integration task, the relative difference of evidence for the two classes, that is evidence for class one minus evidence for class two, are all integers in the range $[-T, T]$).

### Simple integration task with delay

A modified version of the simple integration task outlined above involves adding a delay period between the last of the stimulus inputs and the go signal (*Figure 2c*). We denote the length of the delay period by $T_{\text{delay}}$ and $T_{\text{delay}} < T - 1$. During the delay period, the sequence inputs (without noise) are simply the zero vector, $\mathbf{x}_t = \mathbf{0}$ for $t = T - 1 - T_{\text{delay}}, \ldots, T - 1$. Unless otherwise stated, we consider the case of $T = 40$ and $T_{\text{delay}} = 20$ for this task. We briefly explore the effects of training networks on this task where the delay input has a small nonzero magnitude, see *Figure 5—figure supplement 1*.

### Contextual integration task (retrospective and prospective)

For the retrospective context task, we test the network's ability to hold onto multiple pieces of information and then distinguish between said information from a contextual clue (*Figure 7a*). The prospective integration task is the same but has the contextual cue precede the stimulus sequence (*Figure 7—figure supplement 1d*). Specifically, we simultaneously pass the network *two* $N = 2$ simple integration tasks with delay by concatenating their inputs together. The label of the entire sequence is the label of one of the two integration subtasks, determined by the context which is randomly chosen uniformly over the two possibilities (e.g. subtask 1 or subtask 2) for each sequence. Note for each of the subtasks, labels take on the values $\ell = 1, 2$, so there are still only two possible output labels.

As with above, each task has its various inputs mapped to random binary vectors and the full concatenated input has an expected magnitude of 1. We specifically considered the case where the input size was $d = 50$, so each subtask has 25-dimensional input vectors. The full sequence length was $T = 40$, where $T_{\text{stimulus}} = 19$ and $T_{\text{context}} = 5$. For both the retrospective and prospective setups, context was passed with 10 delay time steps before it, and 5 delay time steps after it.

### Continuous integration task

The continuous integration tests the networks ability to integrate over a continuous values, rather than the discrete values used in the pure integration task above (*Figure 7e*).

The continuous input is randomly generated by first determining the mean value, μ, for a given example by drawing uniformly over the range $[-0.1875, 0.1875]$. For a sequence length of $T$, at each time step the continuous signal is drawn from the distribution $\mathcal{N}\left(\mu, T/750\right)$. Each example has a binary label corresponding to whether μ is positive or negative. Numerical values were chosen such that the continuous integration has a similar difficulty to the integration task investigated in *Mante et al., 2013*. The continuous signal is then multiplied by some random binary vector. Unlike the previous tasks, since the continuous input can be negative, the input values to the network can be negative as well. The go signal is still some random binary vector. We specifically consider the case of $T = 20$.

### True-anti contextual integration task

The true-anti contextual integration task is the same as the simple $N = 2$ integration task, except the correct label to a given example may be the class with the *least* evidence, determined by a contextual clue. The contextual clue is uniformly drawn from the set $\{\text{true}, \text{anti}\}$. Each possible clue again one-to-one maps to a random binary vector that is *added* to the random binary vectors of the normal stimulus input at all time steps (*Figure 7—figure supplement 1a*). We specifically consider the case of $T = 20$.

Note this task is not discussed in detail in the main text, instead the details are shown in *Figure 7—figure supplement 1*. We find both the MPN and GRU are easily able to learn to solve this task.

## NeuroGym tasks

We also train our networks on 19 of the supervised learning tasks of the NeuroGym package (*Molano-Mazon et al., 2022*). As mentioned in the main text, unlike the other tasks considered in this work, the NeuroGym tasks require the network to output responses at all time steps of a sequence rather than following a go signal. Additionally, sequences of the NeuroGym tasks often contain several different trials of a given task, meaning the networks do not automatically have their states reset at the beginning of each distinct task trial. All NeuroGym tasks use default parameters and a sequence length of $T = 100$. See (*Molano-Mazon et al., 2022*) for details about each of the tasks we consider.

We pass all inputs of the NeuroGym tasks through a fixed random linear layer. This is because some tasks require the networks to respond during delay periods represented by zero input and with zero input into MPN and MPNpre they cannot give a modulation-dependent response. Specifically, for a NeuroGym input size $\mathbf{x}'_t \in \mathbb{R}^{d'}$, we generate the random matrix $\mathbf{W}_{\text{rand}} \in \mathbb{R}^{d \times d'}$ and random vector $\mathbf{b}_{\text{rand}} \in \mathbb{R}^d$ using Xavier initialization (see below). All NeuroGym inputs are passed through this matrix to yield the inputs to the network via

$$\mathbf{x}_t = \mathbf{W}_{\text{rand}} \mathbf{x}'_t + \mathbf{b}_{\text{rand}} \,. \tag{9}$$

The values of $\mathbf{W}_{\text{rand}}$ and $\mathbf{b}_{\text{rand}}$ are fixed during training and are redrawn for each separate initialization. We use $d = 10$ across all NeuroGym tasks.

## Networks

In this section, we give a more thorough description for the various networks consider in this work. The number of adjustable parameters in the networks as well as the scaling of the number of state update operations they require are given in *Table 1*. Unless otherwise stated, throughout this work we take networks to have input and hidden neuron counts of $d = 50$ and $n = 100$, respectively. In this setting, the MPN has roughly 1/3 the number of trainable parameters as the Vanilla RNN and 1/9 the number of the GRU.

## Multi-plasticity network

The multi-plasticity networks used in this work can be thought of as a generalization of a two-layer, fully-connected, feedforward network, given by

$$\mathbf{h}_t = \phi \left( \mathbf{W}_{\text{inp}} \mathbf{x}_t + \mathbf{b} \right) \,, \tag{10a}$$

$$\mathbf{y}_t = \mathbf{W}_{\text{RO}} \mathbf{h}_t + \mathbf{b}_{\text{RO}} \,, \tag{10b}$$

where $\mathbf{W}_{\text{inp}} \in \mathbb{R}^{n \times d}$ are the weights connecting the input and hidden layer, $\mathbf{W}_{\text{RO}} \in \mathbb{R}^{N \times n}$ are the weights connecting the hidden and output layer, and $\mathbf{h}_t \in \mathbb{R}^n$ are the hidden activities. The function $\phi \left( \cdot \right)$ is some activation function, usually non-linear, which is applied element-wise. Weights $\mathbf{W}_{\text{inp}}$, $\mathbf{W}_{\text{RO}}$, $\mathbf{b}$, and $\mathbf{b}_{\text{RO}}$ are adjusted during training (see below for details).

The primary difference between the MPN and the above network is that the weights between the input and hidden layers are modified by a time-dependent synaptic modulation (SM) matrix, $\mathbf{M}_t$. That is, *Equation 10a* is replaced by

$$\mathbf{h}_t = \phi \left( \left( \mathbf{M}_{t-1} \odot \mathbf{W}_{\text{inp}} \right) \mathbf{x}_t + \mathbf{W}_{\text{inp}} \mathbf{x}_t + \mathbf{b} \right) \,, \tag{11a}$$

$$= \phi \left( \left[ \left( \mathbf{M}_{t-1} + \mathbf{1} \right) \odot \mathbf{W}_{\text{inp}} \right] \mathbf{x}_t + \mathbf{b} \right) \,, \tag{11b}$$

where $\mathbf{M}_{t-1} \in \mathbb{R}^{n \times d}$, the same dimensions as $\mathbf{W}_{\text{inp}}$, and $\mathbf{1} \in \mathbb{R}^{n \times d}$ has 1 for all its elements. Here, '$\odot$' represent an element-wise multiplication of the two matrices (the Hadamard product). In the main text, we set $\mathbf{b} = \mathbf{0}$ and $\phi \left( \cdot \right) = \tanh \left( \cdot \right)$. The former condition we take this to represent the lack of a background signal, and does not change the qualitative results of the main text. In practice, no significant difference in dynamics from the sigmoid activation function was observed, and the tanh was chosen to better match the computational structure of the RNNs and existing literature on their

dynamics. In *Figure 3—figure supplements 1e–h–3*, we show the dynamics are qualitatively the same for $\phi\left(\cdot\right) = \max\left(\cdot, 0\right)$, i.e. the ReLU activation function.

The SM matrix, $\mathbf{M}_t$, serves as an internal state of the network, intialized at $\mathbf{M}_0 = \mathbf{0} \in \mathbb{R}^{n \times d}$. As mentioned in the main text, throughout this work we consider two types of SM matrix updates intended to broadly model synaptic modulations that occur over distinct timescales. The primary mechanism we investigate uses both the pre- and postsynaptic firing rates,

$$\mathbf{M}_t \quad = \lambda \mathbf{M}_{t-1} + \eta \mathbf{h}_t \mathbf{x}_t^T , \tag{12}$$

where $\eta$ and $\lambda$ are parameters that can be adjusted during training. The alternative modulation update is only dependent upon the presynaptic firing rate,

$$\mathbf{M}_t \quad = \lambda \mathbf{M}_{t-1} + \eta \frac{\mathbf{1}}{\sqrt{n}} \mathbf{x}_t^T , \tag{13}$$

where $\mathbf{1} \in \mathbb{R}^n$ is the all 1s vector. Here, the $1/\sqrt{n}$ factor ensures the $\mathbb{R}^n$ vector in the outer product has $\mathcal{O}(1)$ magnitude. As mentioned in the main text, we refer to networks with the synaptic modulation update of *Equation 13* as the *MPNpre*, reserving just *MPN* to refer to the more general associative update of *Equation 12*.

Notably, unlike the weight modification from backpropagation, the SM matrix weight modification is *local*. That is, the information to update a given synapse/weight only comes from nodes to which the weight is directly connected. Note the convention we have chosen for the time labels in these update expressions means, in order for $\mathbf{h}_1$ to be well defined, we must specify an initial state for $\mathbf{M}_0$, which throughout this work we take to be $\mathbf{M}_0 = \mathbf{0}$. It is possible to train $\mathbf{M}_0$ as an additional set of parameters for the network. In practice, we don't observe significant qualitative differences in the dynamics when this is done.

We let $\eta \in \mathbb{R}$, while $\lambda$ must obey $0 \leq \lambda \leq \lambda^{\max}$ where $\lambda^{\max}$ is a hyperparameter of the MPN. Throughout this work we choose $\lambda^{\max} = 0.95$ unless otherwise noted. This is to enforce a level of biological realism, where the SM matrix exponentially decays. In practice, during training we find that $\lambda$ comes close to saturating $\lambda^{\max}$. Note for $\lambda = 0.95$, in the usual settings where we train for 20 or 40 sequence steps, this means by the time the go signal is passed the first input perturbations decay to roughly 0.35 and 0.13 of their initial values, respectively. As mentioned in the main text, we observe no major difference in performance or dynamics for networks that learn positive or negative $\eta$ (see, for example, *Figure 6—figure supplement 1f*). Although (*Tyulmankov et al., 2022*) finds performance advantages for $\eta < 0$, we note that our network is different in that it is more symmetric with respect to signs flips (the only asymmetry is introduced by the input signal). In particular, (*Tyulmankov et al., 2022*) uses sigmoid activations that limit firing rates to be positive whereas here we use hyperbolic tangent.

The output of the network is given by the expression *Equation 10b*. For ease of interpretation of the mapping between hidden activity and the network output, throughout the main text we take $\mathbf{b}_{\mathrm{RO}} = \mathbf{0}$. Once more, dynamics and quantiative results do not differ significantly from this change.

To emphasize its functional dependence on only the current input and the synaptic modulations, the MPN can be written in the generic form $\mathbf{M}_t = G\left(\mathbf{M}_{t-1}, \mathbf{x}_t\right)$. Similarly, the hidden layer activity is given by the generic expression $\mathbf{h}_t = G'\left(\mathbf{M}_{t-1}, \mathbf{x}_t\right)$, which makes it fundamentally different than the RNNs' generic expression, $\mathbf{h}_t = F\left(\mathbf{h}_{t-1}, \mathbf{x}_t\right)$, see below for additional details.

To understand the dynamics of the MPN, it is occasionally useful to see what the network's hidden activity would look like *if* the go signal were passed to it at the current time step. This is done by translating any given state $\mathbf{M}$ into its corresponding post-go-signal hidden activity by doing what we call a *go signal projection*, defined by

$$\mathbf{h}^{\mathrm{GO}}\left(\mathbf{M}\right) \quad \equiv \phi\left(\left(\mathbf{M} \odot \mathbf{W}_{\mathrm{inp}}\right) \mathbf{x}^{\mathrm{GO}} + \mathbf{W}_{\mathrm{inp}} \mathbf{x}^{\mathrm{GO}} + \mathbf{b}\right) , \tag{14}$$

where $\mathbf{x}^{\mathrm{GO}}$ is the go signal input. That is, this defines a mapping from any MPN state to a hidden activity. An example of this projection is shown in *Figure 3—figure supplement 1c, d*, corresponding to the MPN trained on the $N = 2$ simple integration task discussed in *Figure 3*.

## Additive multi-plasticity network

An alternative model for the MPN, where the SM matrix additively modifies the input weights, is given by *Tyulmankov et al., 2022*

$$\mathbf{h}_t = \phi\left(\left(\mathbf{M}_{t-1} + \mathbf{W}_{\text{inp}}\right)\mathbf{x}_t + \mathbf{b}\right), \tag{15}$$

where once more $\mathbf{M}_{t-1} \in \mathbb{R}^{n \times d}$. Other than the above expression for hidden neuron activity, this network is identical to the MPN with element-wise multiplication above. Tests on the performance of the additive MPN compared to the multiplicative MPN used throughout this work found the additive model to generally perform worse than its multiplicative counterpart (*Figure 6—figure supplement 1*). (We note that the additive MPN setup used here is slightly different than the 'HebbFF' used in *Tyulmankov et al., 2022*. Like the MPN used throughout this work we add no bias to the hidden or output layers, use a tanh activation function, initialize parameters (including $\eta$) using Xavier initialization, limit $\lambda$ to $\lambda^{\text{max}} = 0.95$, and train with L1 regularization. We also note the SM matrix is denoted by $\mathbf{A}$ instead of $\mathbf{M}$ in *Tyulmankov et al., 2022*). The dynamics of this network are also found to be low-dimensional and are shown in *Figure 3—figure supplements 1i-l–3*. An in-depth analysis of its dynamics is outside the scope of this work.

## Postsynatpic $\mathbf{M}_t$ updates

Since throughout this work we consider a modulation updates that are dependent upon either *both* the pre- and postsynaptic firing rates or only the presynaptic firing rate, for completeness here we also consider the SM matrix update that is only postsynaptic dependent. Specifically, we consider an update of the form,

$$\mathbf{M}_t = \lambda\mathbf{M}_{t-1} + \eta\mathbf{h}_t\frac{\mathbf{1}_d^T}{\sqrt{d}}, \tag{16}$$

where $\mathbf{1}_d$ is the d-dimensional vectors of all 1's. That is, the explicit dependence on the pre- or postsynaptic firing rates of *Equation 12* is once again just replaced by a uniform vector. The restriction of information does not appear to change the ability of the network to learn significantly, for only postsynaptic dependence we find the network achieve $98.8 \pm 1.2\%$ accuracy, compared to a baselines of $99.7 \pm 0.6\%$ and $98.0 \pm 1.3\%$ for networks with the associative or presynaptic update rule (mean±s.e., across 10 initializations).

## Recurrent neural networks

Throughout this work, we will compare the MPN to RNNs that are common artificial neural networks used in both the neuroscience and machine learning communities. The simplest version of the RNN is the Vanilla RNN, which can again be thought of as a simple modification to the two-layer feedforward network, *Equation 10a*, *Equation 10b*. Now the hidden layer neurons serve as an additional input to the next layer,

$$\mathbf{h}_t = \phi\left(\mathbf{W}_{\text{rec}}\mathbf{h}_{t-1} + \mathbf{W}_{\text{inp}}\mathbf{x}_t + \mathbf{b}\right), \tag{17a}$$

$$\mathbf{y}_t = \mathbf{W}_{\text{RO}}\mathbf{h}_t + \mathbf{b}_{\text{RO}}, \tag{17b}$$

where $\mathbf{W}_{\text{rec}} \in \mathbb{R}^{n \times n}$ are weights trained in the same manner as those in $\mathbf{W}_{\text{inp}}$. Note with this convention for $\mathbf{h}_1$ to be well defined we must specify the initial state $\mathbf{h}_0$, which we always take to be $\mathbf{h}_0 = \mathbf{0} \in \mathbb{R}^n$. Similar to the MPNs above, $\mathbf{h}_0$ could be trained as an additional parameter, but we simply fix it in this work. Additionally, like the MPN, throughout this work we take $\phi\left(\cdot\right) = \tanh\left(\cdot\right)$ and $\mathbf{b}_{\text{RO}} = \mathbf{0}$.

Vanilla RNNs suffer from the vanishing/exploding gradient problem and more sophisticated units have been introduced to remedy these problems. Perhaps one of the two most famous generalizations of the Vanilla RNN is the Gated Recurrent Unit (GRU) (*Cho et al., 2014*). The GRU introduces several gates to the Vanilla RNN's hidden activity update expressions. These gates control how the hidden activity of the network is changed from one time step to the next via *additive* updates, allowing information to persist for longer timescales (*Hochreiter and Schmidhuber, 1997*; *Ba et al., 2016*). The explicit expressions for the GRU are given by

$$\mathbf{u}_t = \sigma\left(\mathbf{W}_{\text{rec}}^{\text{u}}\mathbf{h}_{t-1} + \mathbf{W}_{\text{inp}}^{\text{u}}\mathbf{x}_t + \mathbf{b}^{\text{u}}\right), \tag{18a}$$

$$\mathbf{r}_t \quad = \sigma\left(\mathbf{W}_{\text{rec}}^{\text{r}}\mathbf{h}_{t-1} + \mathbf{W}_{\text{inp}}^{\text{r}}\mathbf{x}_t + \mathbf{b}^{\text{r}}\right),$$ (18b)

$$\hat{\mathbf{h}}_t \quad = \phi\left(\mathbf{W}_{\text{rec}}\left(\mathbf{h}_{t-1} \odot \mathbf{r}_t\right) + \mathbf{W}_{\text{inp}}\,\mathbf{x}_t + \mathbf{b}\right),$$ (18c)

$$\mathbf{h}_t \quad = \left(1 - \mathbf{u}_t\right) \odot \hat{\mathbf{h}}_t + \mathbf{u}_t \odot \mathbf{h}_{t-1},$$ (18d)

where $\mathbf{u}_t$ and $\mathbf{r}_t$ are the update and reset gate vectors, respectively. The hidden activity is translated to an output through a readout layer, identical to *Equation 17b*.

Both the above RNNs can be written in the generic form $\mathbf{h}_t = F\left(\mathbf{h}_{t-1}, \mathbf{x}_t\right)$. That is, despite its complicated update expressions, the GRUs hidden state at a given time step is still only a function of its current input and previous hidden state. We again note the similarity and differences of this expression to the updates expression for the MPN, $\mathbf{h}_t = G'\left(\mathbf{M}_{t-1}, \mathbf{x}_t\right)$.

## Training

All networks are trained using standard backpropagation (through time), as implemented through the PyTorch package. Networks are trained using ADAM with default parameters (*Kingma and Ba, 2014*) and a constant learning rate, set to $1 \times 10^{-3}$. All trained parameters of the network are subject to L1 regularization, and unless otherwise stated the coefficient of said regularization is $10^{-4}$. In all training settings, Gaussian noise of expected magnitude 0.1 is added to the inputs of the network. Throughout this work, training was conducted for a minimum number of time steps and then stopped under three possible conditions: (1) a (rolling average) validation accuracy was met, (2) the (rolling average) validation loss became saturated, or (3) the maximum training time was reached. See below for details on when each of these thresholds were used. Gradients were clipped to 10 to avoid gradient explosion. All weight matrices are initialized using Xavier initialization, e.g. for $\mathbf{W}_{\text{inp}} \in \mathbb{R}^{n \times d}$ each element is drawn from a uniform distribution over $[-\gamma, \gamma]$ with $\gamma = \sqrt{6/\left(n + d\right)}$.

For the MPN, $\eta$ is similarly drawn from a uniform distribution between $\left[-\sqrt{3}, \sqrt{3}\right]$ is initialized to its maximum value $\lambda^{\text{max}}$, as generally we find it to approach its maximal value during training anyway.

## **Theoretical analysis**

### MPN analysis

Here we give additional details of the theoretical approximations for MPN dynamics. First note that we can write a given $\mathbf{M}_t$ in terms previous inputs, hidden activity, and its initial state

$$
\begin{aligned}
\mathbf{M}_t \quad &= \lambda\mathbf{M}_{t-1} + \eta\mathbf{h}_t\mathbf{x}_t^T, \\
&= \lambda^2\mathbf{M}_{t-2} + \eta\lambda\mathbf{h}_{t-1}\mathbf{x}_{t-1}^T + \eta\mathbf{h}_t\mathbf{x}_t^T, \\
&\vdots \\
&= \eta\sum_{s=0}^{t-1}\lambda^s\mathbf{h}_{t-s}\mathbf{x}_{t-s}^T.
\end{aligned}
$$ (19)

where, in reaching the final line, we have assumed that $\mathbf{M}_0 = \mathbf{0}$.

In the settings we train the MPNs in this work, *we find the effect of the modulation matrix on the hidden activity to be small compared to the unmodulated weights*. That is, in general, the contributions to the hidden activity in *Equation 11a* obey

$$\left\|\left(\mathbf{M}_{t-1} \odot \mathbf{W}_{\text{inp}}\right)\mathbf{x}_t\right\|_2 \ll \left\|\mathbf{W}_{\text{inp}}\mathbf{x}_t + \mathbf{b}\right\|_2.$$ (20)

Thus, the leading-order contributions to the dynamics of both the hidden activity and the SM matrix come from expanding in terms of the number of modulations and neglecting terms with compounded modulation contributions. In Appendix A, we show explicitly why this occurs in the setup we consider in this work. (In brief, writing out the exact expression for $\mathbf{M}_t$, several terms of the form $\left(\mathbf{x} \odot \mathbf{x}'\right)$ appear. By definition of our input vectors, the non-zero components of the vector $\left(\mathbf{x} \odot \mathbf{x}'\right)$ are $\frac{2}{d}$, see *Equation 8*. These components are small compared to the size of terms of a stand-alone input $\mathbf{x}$ (i.e. without the Hadamard product), whose nonzero-components are of size $\sqrt{\frac{2}{d}}$. Thus, any term with $\left(\mathbf{x} \odot \mathbf{x}'\right)$ is small compared to a term with just $\mathbf{x}$). Given the relative size of terms discussed above, the leading-order hidden activity can then be approximated by

$$\mathbf{h}_t \approx \phi\left(\mathbf{W}_{\text{inp}}\mathbf{x}_t + \mathbf{b}\right), \tag{21}$$

which is just the expression of the hidden activity in an unmodulated feedforward network. Plugging this approximation into the update expression for $\mathbf{M}_t$ above, we arrive at an approximation for the modulation matrix

$$
\begin{aligned}
\mathbf{M}_t &\approx \eta \sum_{s=0}^{t-1} \lambda^s \phi\left(\mathbf{W}_{\text{inp}}\mathbf{x}_{t-s} + \mathbf{b}\right) \mathbf{x}_{t-s}^T, \\
&\approx \lambda\mathbf{M}_{t-1} + \eta\phi\left(\mathbf{W}_{\text{inp}}\mathbf{x}_t + \mathbf{b}\right)\mathbf{x}_t^T.
\end{aligned}
\tag{22}
$$

Plugging in $\phi = \tanh$ and $\mathbf{b} = \mathbf{0}$, this matches the approximation used in *Equation 5* of the main text (again see Appendix A for additional details).

Notably, the leading-order expression for $\mathbf{h}_t$ above does not capture the variation due to accumulated evidence we see in the hidden activity. Although subleading, such variation is important for the MPN to solve the integration task. To get a more accurate approximation, we keep terms of subleading order, i.e. terms that arise from a single application of the SM matrix. In this case we arrive at the approximation for hidden activity used in the main text,

$$\mathbf{h}_t \approx \phi\left(\mathbf{W}_{\text{inp}}\mathbf{x}_t + \mathbf{b} + \eta\sum_{s=1}^{t-1}\phi\left(\mathbf{W}_{\text{inp}}\mathbf{x}_{t-s} + \mathbf{b}\right)\odot\left(\mathbf{W}_{\text{inp}}\left(\mathbf{x}_t\odot\mathbf{x}_{t-s}\right)\right)\right). \tag{23}$$

Once more, details of how one arrives at this expression are given in Appendix A. One can of course continue expanding in terms of the modulation matrix to get increasingly accurate and complicated expressions. Notably, from the above expression, one can see that in order for the SM matrix to have a nonzero contribution requires the input of the current and previous time steps to have a nonzero Hadamard product. This is especially important for the final hidden activity, which in this work we take to be the go signal, otherwise all subleading contributions to the hidden activity are zero and the network cannot train. This would occur if one used one-hot inputs for all distinct inputs to the network.

## MPNpre analysis

The lack of hidden activity dependence significantly simplifies the expressions for the MPNpre. In particular, $\mathbf{M}_t$ in terms previous inputs is simply given by

$$\mathbf{M}_t = \frac{\eta}{\sqrt{n}}\sum_{s=0}^{t-1}\lambda^s \mathbf{1}\mathbf{x}_{t-s}^T. \tag{24}$$

where once again we have assumed that $\mathbf{M}_0 = \mathbf{0}$. Note this is the same expression as the MPN, where $\mathbf{h}_{t-s}$ has been replaced by $\mathbf{1}/\sqrt{n}$. However, the lack of hidden activity dependence in this expression means the recurrent expression for $\mathbf{h}_t$ is no longer dependent upon previous hidden activities. As such, the *exact* expression for the hidden activity in terms of previous inputs is given by

$$\mathbf{h}_t = \phi\left(\mathbf{W}_{\text{inp}}\mathbf{x}_t + \mathbf{b} + \frac{\eta}{\sqrt{n}}\sum_{s=1}^{t-1}\mathbf{W}_{\text{inp}}\left(\mathbf{x}_t\odot\mathbf{x}_{t-s}\right)\right). \tag{25}$$

This is the same as the expression for the MPN above, *Equation 23*, except there is now no need to approximate the explicit $\mathbf{h}_{t-s}$ dependence.

## Review of RNN analysis

Since we have observed RNNs tend to operate in close vicinity to attractors that in our case are slow/fixed points, we are motivated to approximate their behavior using a linear expansion (*Maheswaranathan et al., 2019a*). If we expand the generic RNN expression, $F\left(\mathbf{h}, \mathbf{x}\right)$, to linear order about the arbitrary location $\left(\mathbf{h}, \mathbf{x}\right) = \left(\mathbf{h}^e, \mathbf{x}^e\right)$, we have

$$\mathbf{h}_t \approx F\left(\mathbf{h}^e, \mathbf{x}^e\right) + \mathbf{J}^{\text{rec}}\big|_{\left(\mathbf{h}^e, \mathbf{x}^e\right)}\left(\mathbf{h}_{t-1} - \mathbf{h}^e\right) + \mathbf{J}^{\text{inp}}\big|_{\left(\mathbf{h}^e, \mathbf{x}^e\right)}\left(\mathbf{x}_t - \mathbf{x}^e\right), \tag{26}$$

where we have defined the Jacobian matrices

$$J_{ij}^{\text{rec}}\left(\mathbf{h}, \mathbf{x}\right) \equiv \frac{\partial F(\mathbf{h},\mathbf{x})_i}{\partial h_j}, \qquad J_{iJ}^{\text{inp}}\left(\mathbf{h}, \mathbf{x}\right) \equiv \frac{\partial F(\mathbf{h},\mathbf{x})_i}{\partial x_J}. \tag{27}$$

If we take $\mathbf{h}^{\text{e}}$ to be a slow/fixed point under the input $\mathbf{x}^{\text{e}} = \mathbf{0}$, we have that $\mathbf{h}^{\text{e}} \approx F\left(\mathbf{h}^{\text{e}}, \mathbf{x}^{\text{e}} = \mathbf{0}\right)$. Inserting this into the above expression, we can approximate the effect of a given input at time $\mathbf{x}_t$ on the hidden activity $\mathbf{h}_{s+t}$ by $\left(\mathbf{J}^{\text{rec}}\right)^s \mathbf{J}^{\text{inp}}\mathbf{x}_t$. Thus an approximation of the state at time $t$, assuming $\mathbf{h}_0 = \mathbf{0}$, is given by *Maheswaranathan et al., 2019a*

$$\mathbf{h}_t \approx \sum_{s=0}^{t} \left(\mathbf{J}^{\text{rec}}\right)^{t-s} \mathbf{J}^{\text{inp}}\mathbf{x}_s. \tag{28}$$

Using SVD, we can always write $\mathbf{J}^{\text{rec}} = \mathbf{R}\mathbf{\Lambda}\mathbf{L}$ with $\mathbf{R} = \mathbf{L}^{-1}$ and $\mathbf{\Lambda} = \text{diag}\left(\lambda_1, \ldots, \lambda_n\right)$. Then this reduces to

$$\begin{aligned} \mathbf{h}_t &\approx \sum_{s=0}^{t} \mathbf{R}\mathbf{\Lambda}^{t-s}\mathbf{L}\mathbf{W}_{\text{inp}}\mathbf{x}_s, \\ &= \sum_{s=0}^{t}\sum_{i=1}^{n} \mathbf{r}_i \lambda_i^{t-s}\mathbf{l}_i^T\mathbf{W}_{\text{inp}}\mathbf{x}_s, \end{aligned} \tag{29}$$

where in the second line $\mathbf{r}_i$ and $\mathbf{l}_i$ are the columns of $\mathbf{R}$ and $\mathbf{L}$, respectively. From this expression it is straightforward to understand the impact each input will have on a given hidden state in terms of its projection onto the eigenmodes. We see that an eigenmode's contribution will disappear/explode over time if $\lambda < 1$ or $\lambda > 1$, respectively. In practice, trained networks tend to have one eigenvalue $\lambda \approx 1$ for each integration dimension (e.g. a single eigenvalue at $\lambda \approx 1$ for a line attractor) (*Aitken et al., 2020*). So long as there are integrator modes, we see all inputs have the ability to contribute to a given $\mathbf{h}_t$ on roughly equal footing. This is fundamentally different from the operation of the MPN discussed above.

## Network dynamics, dimensional reduction, and modulation bounds

### Determining dimensionality, PCA projection, and explained variance

Throughout this work, we use PCA to visualized the dynamics. This is always done by passing some test set to the network of size $m$, and collecting all activity vectors/matrices (e.g. hidden activity or SM matrices) over the entire test set and some time period, $S$. Unless otherwise stated, this time period is over the entire input sequence, that is $S = T$, and thus this yields on order $m \times T$ vectors/matrices. Let $D$ denote the size of the space said vectors or matrices live in (e.g. $D = n$ for hidden activity, $D = n \times d$ for SM matrices). PCA is then performed over this set, which yields some set of PCA vectors, $\mathbf{w}_\alpha$, and their associated ratio of variance explained, $v_\alpha$, for $\alpha = 1, \ldots, \min\left(D, m \times T\right)$. The dimensionality of activity is determined by calculating the *participation ratio* on the $v_\alpha$ (*Aitken et al., 2020*),

$$\text{PR} \equiv \frac{\left(\sum_{\alpha=1}^{D} v_\alpha\right)^2}{\sum_{\alpha=1}^{D} v_\alpha^2}. \tag{30}$$

Except for the smallest networks considered in this work, we always find $\text{PR} \ll D$, which we take to mean the activity operates in a low-dimensional space.

To determine the hidden activity's proportion of variance explained by either the accumulated evidence or the present input, we associate each hidden activity vector, $\mathbf{h}_t$, with the relative score difference between the two classes and present input, $\mathbf{x}_t$, at the given time step. We then use scikit-learn's LinearRegression class with default parameters to fit the hidden activity over an entire test to the accumulated evidence and present input individually. Note for the former, which consists of four possible input categories, we use three indicator functions with the reference case chosen to be the 'null' input. A linear fit was chosen for simplicity and the authors felt it was appropriate given the fact the outputs of the network are also a linear function of the hidden activity. The reported proportion of variance explained correspond to the $r^2$ of the individual fits, averaged over 10 separate initializations.

### MPNpre dynamics

For the MPNpre, using the update rule that is only dependent upon the presynaptic firing rates does not considerably change the dynamics that we observed for the MPN's associative learning rule. The

hidden activity is again separated into distinct input clusters that each vary with accumulated evidence (*Figure 3—figure supplement 1e and f*). The SM matrix activity grows as the sequence is read in and also encodes the accumulated evidence of the phrase (*Figure 3—figure supplement 1g and h*).

Comparing the hidden activity expressions for the MPN and MPNpre, *Equations 23 and 25*, we can explicitly see a manifestation of the simpler modulation updates: the MPN's modulations that are dependent upon both the pre- and postsynatpic activity means the modulations are modified via a more complex dependence on previous inputs. Indeed, the MPNpre has significantly less control over the form of its modulations, the only part of the update rule that can be modified through training is $\eta$. Meanwhile, the MPN can modify $\mathbf{W}_{\mathrm{inp}}$ which in turn affects the $\mathbf{h}_t$, and thus can have significantly more nuanced modulation updates. We see this explicitly when comparing the accumulated evidence variation of the input clusters in the two networks. Comparing the insets of *Figure 3f* and *Figure 3—figure supplement 1g*, we see the alignment of all four input clusters of the MPNpre are high rather than just the go signal cluster for the MPN. Averaging several initializations of the networks, the MPN's readout alignment for words that are not the go signal is $0.16 \pm 0.08$ whereas for the MPNpre it is $0.79 \pm 0.08$ (mean±s.e.). Note this also means the individual input clusters are significantly more aligned with one another than in the case of the MPN. These effects are a direct result of the additional term that modifies the hidden state dependence in *Equation 23* that is absent in *Equation 25*.

## Generalization of MPN dynamics for $N > 2$

In the main text, the dynamics of the MPN was discussed for the case of an $N = 2$ integration task. Here we expand on how said dynamics generalize for $N > 2$. Throughout this discussion, we always assume the number of classes is significantly smaller than the hidden activity dimension, i.e. $N \ll n$.

Like the $N = 2$ case, for $N > 3$ we observe the hidden activity to continue to be low-dimensional and separated into distinct input-clusters (dependent upon the most recent input to the network). *Figure 3—figure supplement 3* shows a visualization of the dynamics for the MPN trained on an $N = 3$ task. Within each input-cluster, the hidden activity continues to vary with accumulated evidence. However, since keeping track of the relative scores between classes now requires the network to hold more than a single number, the variance of accumulated evidence within each input-cluster takes on a more complex structure. For example, in the $N = 3$ case, we see each input-cluster is roughly triangle-like, with the hidden activity belonging to the 3 possible output labels clustering toward each of the corners (*Figure 3—figure supplement 3e*). Similar to the RNN dynamics (see below), we expect accumulated evidence for an N-class task to take on the approximate shape of an N - 1-simplex within each input-cluster (*Aitken et al., 2020*).

Once again, the SM matrix continues to monotonically evolve along a particular direction with sequence index (*Figure 3—figure supplement 3g*). We also find the SM matrix again encodes accumulated evidence of the distinct classes (*Figure 3—figure supplement 3h*). That is, each distinct input deflects the SM matrix in a distinct direction, allowing said matrix to encode all previous inputs to the MPN up to that point.

As mentioned in the main text, we limit our analysis of dynamics to the case where the separate classes of the task are uncorrelated. Similar to RNNs, we expect the dynamics to qualitatively change in cases where classes are non-uniformly correlated with one another (*Aitken et al., 2020*).

## Review of RNN dynamics

Recall that to solve the N-class simple integration task, at a minimum a network needs to keep track of $N - 1$ relative evidence values (see Sec. 5.1 above). In general, so long as $n \ll N$, after training the RNNs' hidden activity lies in a low-dimensional subspace that has the approximate shape of an $N - 1$ regular simplex (which has dimension $N - 1$) (*Aitken et al., 2020*). By moving around in said subspace, the RNN encodes the $N - 1$ relative evidence values needed to solve the task. If instead the network needed to keep track of the absolute evidence for each class, we would expect the network's hidden activity to lie in an $N$-dimensional subspace. For example, in the $N = 2$ case, the relative evidence is a single number, so we only see a one-dimensional line attractor (a one-simplex). Beyond two-class, the attractor becomes higher-dimensional, for example $N = 3$ yields a two-dimensional triangular attractor (a two-simplex) because the network needs to keep track of the relative evidence of two distinct pairs of classes, from which it also encodes the relative evidence of the final pair of classes (*Figure 3—figure supplement 3*).

## Modulation bounds

To remove the biologically implausibility of arbitrary large modulations, we briefly investigate the effects of bounding the size of the **M** matrix explicitly. Note bounds on the size of the modulations are usually implicitly implemented by restricting the magnitude of $\lambda$ to $\lambda^{\max}$. For the explicit bounds, all elements of the matrix are restricted to $|M_{iI}| \leq m_b$, where $m_b$ is the size of the modulation bounds. Note that due to the multiplicative modulation expression we use in this work, this means that the total range a given synaptic connection can be modulated depends on the size of the corresponding element of $\mathbf{W}_{\mathrm{inp}}$. For example, if $m_b = 1$, then the total synaptic strength can be anywhere in the range from zero to double its value at the start of the sequence, since

$$0 \leq M_{iI,t} W_{\mathrm{inp},iI} + W_{\mathrm{inp},iI} \leq 2 W_{\mathrm{inp},iI} \,. \tag{31}$$

Biologically, STDP that potentiates synapses up to ~200% of their original values and depresses by up to ~30% at timescales of 10s of minutes has been seen in a wide variety of studies in various parts of the brain (*Sjöström and Häusser, 2006*; *McFarlan et al., 2023*). Other specific studies sometimes find values that are a bit more extreme, for example depression as low as 20% in *Cho et al., 2000*. Over shorter timescales, STSP has been shown to facilitate to a similar relative magnitude, and seems to be capable of almost complete depression (*Campagnola et al., 2022*).

We notice very little to no significant drop in the network's ability to do integration-based tasks when the aforementioned biologically realistic bounds on modulations are introduced. For a two-class integration task the unbounded network achieves $99.8 \pm 0.4\%$ accuracy, while the for $m_b = 1$ and 0.1 we observe $99.7 \pm 0.9\%$ and $96.7 \pm 2.1\%$, respectively (mean±s.e.). Note the latter of these is quite prohibitive, only allowing the synapses to change by 10% of their original value. As expected, the bounds on the modulation matrix result in an overall smaller modulations (*Figure 3—figure supplement 2e*). As a result of the smaller modulation range, we also observe that the hidden activity is also smaller (*Figure 3—figure supplement 2f*) and that the network's learned |$\eta$| is smaller for more restrictive modulation bounds (*Figure 3—figure supplement 2g*). We hypothesize that the smaller hidden activity and learning rates help facilitate the discrimination of modulation differences when they are restricted to smaller values.

Even for the very restrictive case of $m_b = 0.1$, we observe qualitatively similar hidden activity dynamics to the unbounded case discussed in the main text. That is, the hidden activity is separated into distinct input clusters that each encode accumulated evidence at subleading order (*Figure 3—figure supplement 2h*). Meanwhile, the SM matrix activity again grows in time as the sequence is read in (*Figure 3—figure supplement 2i*) and also encodes the accumulated evidence along a perpendicular axis (*Figure 3—figure supplement 2j*). Note the small size of both the hidden activity and SM matrix activity relative to their unbounded counterparts in the main text (*Figure 3*).

## Figure details

### *Figure 1* details
Explicit expressions for the MPN, fully connected network, and (Vanilla) RNN are given above in Sec. 5.2. The details of the backpropagation modified weights are given in Sec. 5.2.3. The details of the modulated synapses/weights are given in *Equation 2a*.

### *Figure 2* details
Details of the simple integration task shown in *Figure 2* can be found in Sec. 5.1 above.

### *Figure 3* details
All subplots in this figure are for networks trained on a two-class integration task with $T = 20$ and no delay period. For this task, there are only four possible inputs to the networks, 'evid$_1$', 'evid$_2$', 'null', or 'go'. Networks are trained until an accuracy threshold of 98% is reached, with a minimum of 2000 training batches.

*Figure 3a and d* show the activity over the input sequence for a few randomly chosen hidden neurons of a single test example. The single test example is randomly chosen from the set of examples that have at least half the of the maximum possible accumulated evidence difference. A large collection of plots in this work are projected into PCA space as a means of visualizing the low-dimensional

dynamics. To generate the plot in **Figure 3b**, all hidden neuron activity, $\mathbf{h}_t$ for $t = 1, \ldots, T$ were collected over a test batch of size 1000. Across this entire set of hidden activity, PCA was performed (with a centering of per-feature means). The projection of the hidden neuron activities onto their first two components is plotted in **Figure 3b**. The hidden activity is colored by the relative evidence of the two classes, that is total evidence for the red class minus evidence for the blue class, at the associated time-step. In addition to the hidden activity, two example trajectories and the two readout vectors are projected onto the hidden neuron PCA space as well. **Figure 3c** contains the same hidden activity projection, but now the hidden states are colored by the most recent input passed to the network, that is whether the input corresponded to 'evid$_1$' (red), 'evid$_2$' (blue), 'null' (grey), or 'go' (purple). The inset was computed by calculating the change in the hidden activity, $\mathbf{h}_t - \mathbf{h}_{t-1}$, induced by the input $\mathbf{x}_t$, again projected into the hidden activity PCA space. Once again, these changes in hidden activity are colored by the corresponding input. Additionally, for each possible input, the set of all hidden activity changes was averaged together to form a mean change in activity, and this was also plotted in the inset as a darker line.

For the MPN hidden activity, **Figure 3e and f** were generated in an identical manner to those in **Figure 3b and c** (with the exception of the inset). Note there is no initial hidden activity for the MPN, just an initial SM matrix. The inset of **Figure 3f** was computed by separating all hidden activities into distinct groups by their most recent input. Then, for each group of hidden activities, PCA was performed on the subset. The 'RO alignment' plotted in the inset is the average across all readout vectors of the cosine angle magnitude between the readout vector, $\mathbf{r}$, and the top PC direction, $\mathbf{w}$, i.e. $|\mathbf{r} \cdot \mathbf{w}| / (\|\mathbf{r}\|_2 \|\mathbf{w}\|_2)$.

The MPN state ($\mathbf{M}_t$) activity shown in **Figure 3g and h** is plotted in an analogous manner to the hidden activity for the previous subplots in this figure. That is, we collect all $\mathbf{M}_t$ for $t = 0, \ldots, T$ over the test set, and perform PCA on the *flattened* matrices. We then project the flattened states onto said PCA space and color them in the same manner as the hidden activity in **Figure 3b and e** (relative evidence of the two classes). **Figure 3g and h** differ only in that they show two different PCA directions along the $y$-axis. The final SM matrix for each input sequence was colored in a slightly darker shade (but still colored by relative evidence) for clarity. We note that the time-evolution direction is not always aligned with the second PC direction as it is in this plot – in our tests it was also often aligned with the first PC direction or diagonally oriented in the PC1-PC2 plane. The inset of **Figure 3h** is completely analogous to that in **Figure 3b**, but for the MPN state rather than the RNN state. That is, it shows the change in SM matrix, $\mathbf{M}_t - \mathbf{M}_{t-1}$, colored by the present input, $\mathbf{x}_t$.

### *Figure 4* details

The networks that generated the data in the top row of this figure were all trained on an $N = 2$, $T = 20$ integration task until they reached an average validation accuracy of 98%, with a minimum training size of 2000 batches. MPNs were trained at a variety of $\lambda^{\max}$ values, but in practice we found the MPNs always saturated said value at its maximum so $\lambda = \lambda^{\max}$. After training, the networks were then passed longer sequences (up to $T = 200$) and their states (i.e. $\mathbf{h}_t$ for the RNNs and $\mathbf{M}_t$ for the MPNs) were tracked as a function of $t$. Note the same MPN was used to generate **Figure 4a** as that used to generate the dynamics plots in **Figure 3[d–f]**. Additionally, the same Vanilla RNN was used to generate **Figure 4b** as that used to generate the dynamics plots in **Figure 3[a–c]**. All data in **Figure 4c and d** were averaged over 10 different network initializations. The normalized final state for the MPN/MPNpre and the RNNs are given by

$$\frac{\|\mathbf{M}_{T-1}\|_2}{\sqrt{nd}}, \qquad \frac{\|\mathbf{h}_T\|_2}{\sqrt{n}}. \qquad (32)$$

Note the final state of the MPN/MPNpre is chosen to be $\mathbf{M}_{T-1}$ because $\mathbf{M}_T$ is not used in the calculation of the final output.

For the delay task shown in the bottom row, we train the networks on an $N = 2$ integration-delay task with $T_{\text{delay}} = 20$ and a total sequence length of $T = 40$. For all networks other than the MPNpre, training is stopped when the networks reach an accuracy threshold of 98% with a minimum training size of 2000 batches. We found for the given sizes of $\lambda$, the MPNpre sometimes could not reach the aforementioned accuracy threshold and thus they were trained to a threshold of 95% instead. **Figure 4e and f** are generated analogously to **Figure 4a and b**, with $T_{\text{delay}}$ varied from 10 to 100. The red and

blue points shown in *Figure 4e and f* correspond to the state projections immediately preceding the onset of the delay. Note *Figure 4f* shows the state space of a GRU while *Figure 4b* shows that of a Vanilla RNN, since the latter was not able to train on the integration-delay task. *Figure 4g and h* are generated analogously to *Figure 4c and d*, using the same definition of normalized final states.

## *Figure 5* details

The networks shown in *Figure 5a and b* were trained on an $N = 3$ integration task, with $T = 40$ and $T_{\text{delay}} = 20$. Sample neurons in *Figure 5a* are randomly chosen from a single randomly chosen test example. To calculate the decoding accuracy in *Figure 5b and a* linear SVC was trained on the hidden activity of each of the networks, with each hidden activity labeled by its corresponding input sequence's label. A linear SVC was chosen because the readout vectors also implement flat decision boundaries (although they are piece-wise in the case of the readout vectors). The linear SVC was implemented using scikit-learn's LinearSVC class with default settings except for number of iterations was increased to 100,000 and balanced class weights. Since the number of classes in this example was $N = 3$, the multi-class strategy was all-vs.-one. Additionally, 10 fold cross-validation was used and the results shown are averaged over folds.

*Figure 5c* contains the activity as a function of time of a few randomly chosen components of the hidden activity and SM matrix of an MPN from a randomly chosen test example. The theoretical predictions come from *Equations 4 and 5*. In *Figure 3d*, to quantify the error in the theoretical approximation we use the following expressions

$$\text{error}_{\mathbf{h}_t} = \frac{\left\| \mathbf{h}_t^{\text{theory}} - \mathbf{h}_t \right\|_2}{\|\mathbf{h}_t\|_2}, \qquad \text{error}_{\mathbf{M}_t} = \frac{\left\| \mathbf{M}_t^{\text{theory}} - \mathbf{M}_t \right\|_2}{\|\mathbf{M}_t\|_2}, \tag{33}$$

where $\|\cdot\|_2$ is the L2-normalization of the correspond vector or matrix (the Frobenius norm) and $\mathbf{h}_t^{\text{theory}}$ and $\mathbf{M}_t^{\text{theory}}$ are the approximations in *Equations 4 and 5*, respectively. See the derivation leading up to *Equations 22 and 23* to see details of these approximations.

## *Figure 6* details

Network capacity in *Figure 6a* was measured by training networks on tasks with an increasingly large number of classes, $N$. A sequence length of $T = 20$ was still used for all $N$. Training was conducted until the rolling average validation loss plateaued. The rolling average validation loss is the average of the last 10 measured validation losses. Since validation loss is only computed 10 batches, this represents an average over the previous 100 batches of training. In particular, if the current rolling average validation loss was larger than the rolling validation loss measured 100 batches prior, the network's training was determined to have plateaued. Occasionally this metric may have caused a network to end training early, hence for the aggregate measure we plotted the median accuracy over 10 separate initializations. In *Figure 6b*, network capacity as a function of sequence length, $T$, was computed in an identical manner. A class size of $N = 2$ was used for all $T$.

For *Figure 6c*, we trained networks at a fixed ratio of (expected) input noise and signal magnitude. Specifically, if the input without noise is $\tilde{\mathbf{x}}$ and the noise is $\mathbf{n}$, then

$$\alpha \equiv \frac{\mathbb{E}\|\tilde{\mathbf{x}}\|_2}{\mathbb{E}\|\mathbf{n}\|_2}. \tag{34}$$

$\alpha = 10$ was chosen for the initial training. We then systematically scanned over $\alpha$ values and measured accuracy (without retraining on different $\alpha$). It was observed that training the networks at lower $\alpha$ did not change the results significantly.

For the echo-state setup in *Figure 6d*, all weights except for the readout matrix, $\mathbf{W}_{\text{RO}}$, were frozen at their initialization values during training. Since in the MPNs, $\lambda$ and $\eta$ play especially important roles in their operation, we initialized $\lambda = \lambda^{\text{max}} = 0.95$ and $\eta = 1.0$. Training was performed until the validation loss plateaued, see above.

*Figure 6e* was generated by first training the networks on an $N = 2$ integration-delay task until an accuracy threshold, measured by the rolling average validation accuracy (over the last 10 measured validation runs), achieves an accuracy of 97% or higher. Then, the network was trained on a new $N = 2$ integration-delay task. Afterwards, we test how much the accuracy on the original $N = 2$ task fell. We take $T = 40$ and $T_{\text{delay}} = 20$ for both tasks. The new task uses the same random vectors for the 'null'

and go signal inputs, but generates new random vectors for the two types of evidence. The label set of the new task is the same as the old task and the readouts must be adjusted to compensate. That is, we use the same readouts for both tasks, so the readouts may get adjusted during the second task. This is chosen to simulate the bottleneck of sparse strong connections between areas of the brain. After the network achieves the same accuracy threshold on the new task, its accuracy on the original task is reported. In this setup, we reduce the networks' batch size to 1 so we the training can be monitored down to individual examples and also reduced the L1 regularization coefficient to $10^{-6}$. Additionally, raw data in *Figure 6e* shows an average over 20 training examples, and some accuracies (for the GRU in particular) are too low to be seen on the plot.

For the MPN, we find the change in dimensionality of the state space (post-training minus pre-training) to be $0.08 \pm 0.04$ (mean±s.e.). Similarly, for the GRU, we find the change to be $0.19 \pm 0.08$ (mean±s.e.). To find the change in GRU line attractor angle, we take the first PC direction of the hidden states to be a good measure of the direction of the line attractor. For both pre- and post-training, we continue to find the dimensionality of the GRU state space to be small, $1.52 \pm 0.10$ and $1.71 \pm 0.13$, respectively (mean±s.e.).

The PC plots in *Figure 6f and g* are generated as per usual for the states of the respective networks (including the novel states). Looking along higher PC directions, we did not observe a large degree of separation between the new class and previous classes in the RNN.

The decoding accuracy in *Figure 6h* was calculating by training a linear SVC on only the final state activity of the MPN and GRU. The sequence labels were used to label the states. Otherwise, the setup for the linear SVC was identical to that used in *Figure 5b*, see above. Since the MPN and GRU have the same $n = 100$ and $d = 50$, this means the state of the MPN has 50 times more components than the GRU. Thus, to ensure the SVCs are being trained on equal footing for the two networks, we dimensionally reduce the (flattened) MPN states using PCA so that said states have the same number of components as the GRU state (100 in this case). Note that in practice, both networks operate their states in significantly smaller subspaces than the number of components they have, and said subspaces are of comparable dimension for the two types of networks (but not the same, and this may contribute to the MPN's higher accuracy).

## *Figure 7* details

Details on the tasks shown in *Figure 7a, e and i* can be found in Sec. 5.1 above. Accuracy on the retrospective task was found by training 10 different initializations on said task with $T = 40$ and $T_{\text{delay}} = 20$ until the average validation loss saturated (with a minimum training time of 2000 batches). In figures for the retrospective integration, we color the example sequences by both their final label as when as their context, yielding $2 \times 2 = 4$ possible colorings.

For *Figure 7[b–d]*, data is shown for a single network, trained to an accuracy threshold of 98%. In *Figure 7d*, we define the output difference as follows. To determine the potential output at any given time-step, we perform the go signal projection of the states $\mathbf{M}_t$, see *Equation 14*, and then further pass this through the readouts,

$$\mathbf{y}_t^{\text{GO}} = \mathbf{W}_{\text{RO}}\mathbf{h}_t^{\text{GO}} . \tag{35}$$

For an entire test set, we then group each sequence of outputs into the $N \times 2 \times N = 2N^2 = 8$ distinct possible combinations of label, subtask, and context. We then compute the average output sequence for each of these eight groupings, to arrive at eight sequences that represent the average output in each situation. At risk of abusing the number of subscripts a variable can take, let $\bar{\mathbf{y}}_{t,\ell,\tau,c}^{\text{GO}}$ be this average output sequence for label $\ell = 1, 2$, subtask $\tau = 1, 2$, and context $c = 1, 2$ at a given time $t$. For $N = 2$, each $\bar{\mathbf{y}}_{t,\ell,\tau,c}^{\text{GO}}$ is a two-dimensional vector. The *readout difference* at a given time $t$, $\Delta y_{t,\tau,c}$, is then defined to be

$$\Delta y_{t,\tau,c} \equiv \tfrac{1}{2}\left(\bar{y}_{1;t,1,\tau,c}^{\text{GO}} - \bar{y}_{2;t,1,\tau,c}^{\text{GO}}\right) + \tfrac{1}{2}\left(\bar{y}_{2;t,2,\tau,c}^{\text{GO}} - \bar{y}_{1;t,2,\tau,c}^{\text{GO}}\right) . \tag{36}$$

Intuitively, this is the average over the output vector-component difference of the two possible labels. For each of these quantities, the network should have learned that this difference should be positive if the sequence is to be labeled correctly (e.g. when the task has label $\ell = 2$, the second component of $\mathbf{y}$ should be larger than the first so $y_2 - y_1 > 0$). The more positive a given $\Delta y_{t,\tau,c}$, the better the

network is distinguishing sequences that belong to the individual labels at the given time step. The four lines in *Figure 7d* are generated by plotting all four combinations of $\tau = 1, 2$ and $c = 1, 2$, where the solid lines correspond to $\tau = c$ and the colors correspond to the $\tau$ label.

*Figure 7g* was generated by randomly choosing an example sequence and a single input and hidden neuron values. Each sequence was then normalized to have zero mean and a maximum magnitude of 1 for better comparison.

*Figure 7j* shows the average accuracy of five separate network initializations for each type of network. All networks were trained on a batch size of 32 for a minimum of 2000 iterations and with hidden biases. Training was ended when either the network achieved a (rolling) accuracy of 99% on the task or the (rolling) validation accuracy saturated. All NeuroGym *Molano-Mazon et al., 2022* used a sequence length of 100 steps and were generated with default parameters. Since the NeuroGym tasks require outputs at all sequence time steps, we compute accuracies across the entire sequence. Additionally, since the NeuroGym tasks are 100 sequence steps long, we set $\lambda^{\max} = 0.99$ for the MPN and MPNpre.

### *Figure 3—figure supplement 1* details
All subplots in this figure were generated in an analogous manner to those in *Figure 3*.

*Figure 3—figure supplement 1a and b* are the same as *Figure 3b and c*, except the network is a GRU instead of a Vanilla RNN. *Figure 3—figure supplement 1c and d* are the go signal projection, see *Equation 14*, of the $\mathbf{M}_t$ states shown in *Figure 3g and h*. *Figure 3—figure supplement 1c* has the states colored by accumulated evidence, *Figure 3—figure supplement 1d* has the states colored by present input. *Figure 3—figure supplement 1e-h* are the same as those in *Figure 3[e–h]*, but for the MPNpre. That is, the SM matrix is updated using only the presynaptic dependent expression in *Equation 13* instead of *Equation 12*. Finally, *Figure 3—figure supplement 1i-l* are also the same as those in *Figure 3[e–h]*, except the MPN has $\phi(\cdot) = \max(0, \cdot)$, that is the ReLU activation function.

### *Figure 3—figure supplement 2* details
*Figure 3—figure supplement 1a-d* are the same as those in *Figure 3[e–h]*, but for the additive MPN of *Equation 15*. See Methods 5.4.5 for a discussion of the results of introducing bounded modulations. The accuracy results and *Figure 3—figure supplements 2e-g* were averaged across 10 separate initializations. Plots *Figure 3—figure supplement 2h, i, j* are generated in an analogous manner to those in *Figure 3e, g and h*.

### *Figure 3—figure supplement 3* details
All subplots in this figure were generated in an analogous manner to those in *Figure 3*, but for the case of $N = 3$ instead of $N = 2$. Hidden activity and SM matrix activity that were previously colored by accumulated evidence are now colored by example label instead.

### *Figure 5—figure supplement 1* details
In *Figure 5—figure supplement 1[a-c]*, we plot several measures of the MPN and GRU trained on the same integration-delay explored in *Figure 5*. For the MPN, we vary the magnitude of the delay input, which throughout the main text was set to 0. The mean hidden activity of *Figure 5—figure supplement 1a* is given by $\bar{h}_t = \frac{1}{n} \sum_{i=1}^{n} |h_{i,t}|$. In *Figure 5—figure supplement 1b*, we again plot the decoding accuracy trained on the hidden activity, see description of *Figure 5b* above. Notably, decoding accuracy during the delay period increases slightly with larger delay input. In *Figure 5—figure supplement 1c*, we plot the time-variation over a rolling time window, normalized relative to the size of activity. The normalized time variability is computed over a $\tau$-time step rolling window. Specifically, for each neuron it is given by

$$\sigma_{i,t}^{\tau} \equiv \frac{\text{Var}\left(\left\{h_{i,t-s}\right\}_{s=0,1,\ldots,\tau-1}\right)}{\frac{1}{\tau} \sum_{s=0}^{\tau-1} h_{i,t-s}^2} , \tag{37}$$

The plot is then created by averaging this quantity over both batches and neurons for all time steps $t \geq \tau$, with $\tau = 5$. We see the MPN not only has significantly higher time-variation than the GRU during the delay period but also the stimulus period that preceded it as well.

*Figure 5—figure supplement 1d* shows the accuracy of analytical approximations, see description of *Figure 5d* above, this time as a function of sequence time. The accuracy of the $\mathbf{h}_t$ approximation gets worse as a function of time due to the compounding effects of the approximation over many time steps.

*Figure 5—figure supplement 1e-g* show the decoding accuracy as a function of decoder train and test time (all use the same color scale shown on the far right). The only difference from the calculation of decoding accuracy from *Figure 5b* is that the train and test times can be different in this setup. *Figure 5—figure supplement 1e* shows the data for a GRU, *Figure 5—figure supplement 1f* for the MPN with zero delay input magnitude, and *Figure 5—figure supplement 1g* for the MPN with a delay magnitude of 0.05.

### Figure 6—figure supplement 1 details

For plots *Figure 6—figure supplement 1a-e*, the MPNpre and additive MPN was trained in an identical setting to the MPN and RNNs of *Figure 6[a–e]*. For the noise plot, the additive MPN was not able to achieve the same accuracy thresholds of the other networks, so training was capped at 16,000 batches. A similar iteration threshold was used for catastrophic forgetting as the networks also occasionally struggled to learn the delay task for certain initializations. For the catastrophic forgetting tasks, the MPNpre had considerable difficulty achieving the 97% accuracy threshold and in some cases reached the maximum training step count (100,000 training samples) without reaching this accuracy threshold. This appears to be an issue with credit assignment in such setups, since it barely achieves better than chance accuracy. In such cases, we omitted these results from the the average behavior, since all other network results were conditioned on reaching the accuracy threshold on both tasks. The MPNpre's worse behavior relative to the MPN on integration tasks with delays was also observed *Figure 6h*. We leave an investigation into such effects for future work. *Figure 6—figure supplement 1j* was generated analogously to *Figure 7j* across 10 initializations of the MPN and MPNpre.

### Figure 7—figure supplement 1 details

Details of the true-anti context task can be found in Sec. 5.1 above. In *Figure 7—figure supplement 1b and c*, we color the SM matrix/hidden activity by both label and context, with the darker colors corresponding to the 'anti-' context. *Figure 7—figure supplement 1b* is the PC projection of the SM matrix over all sequence time steps. *Figure 7—figure supplement 1c* is the PC projection of only the final hidden activity.

The prospective task integration task shown in *Figure 7—figure supplement 1[d-g]* is identical to the retrospective one of the main text and shown in *Figure 7[a–d]*, except that the context comes *before* the evidence sequence (*Figure 7—figure supplement 1d*). Again looking to the MPN's $\mathbf{M}_t$ dynamics, we see the context quickly separates the evidence-less states into two separate clusters (*Figure 7—figure supplement 1e*). Soon after, once evidence starts being passed to the network, the clusters remain distinct and evidence for the individual integration subtasks is accumulated (*Figure 7—figure supplement 1f*). Once again quantifying the information contained in $\mathbf{M}_t$ via the readout difference, we see the subtasks that match the context have their readout difference grow quickly (*Figure 7—figure supplement 1g*). Meanwhile, since the context has already been processed by the network, the evidence needed to solve the irrelevant subtask is stored in a way that is unimportant for the readouts.

*Figure 7—figure supplement 1h and i* quantifies how the accumulated evidence direction aligns with the readout vectors for both the retrospective and prospective contextual integration tasks. The accumulated evidence direction is computed for each subtask and context combination, so four lines are shown in each subplot. At each time step, for each possible subtask/context combination, the SM matrix states are sorted by the amount of accumulated evidence they encode (which takes on a finite number of integer values). All states that have the same accumulated evidence are averaged together, yielding a set of average SM matrix states for all possible accumulated evidence scores at that time step and subtask/context combination. For a given subtask/context combination, the average SM states are mapped to hidden activity using the go signal projection, *Equation 14*, then said hidden activities are fit using PCA, with the top PC direction taken to be the direction of accumulated evidence variation in the hidden activity space. The readout alignment is then the average cosine angle of the top PC direction with the two readout vectors (taking the maximum of the two

possible directions PC1 could point to eliminate ambiguity). Thus, higher readout aligns mean the direction of accumulated evidence variation for the subtask is more aligned with the readout vectors, indicating that the readout vectors are better at distinguishing evidence for the particular subtask. Notably, once context is passed, the irrelevant subtask's accumulated evidence variation becomes close to perpendicular to the readouts, since its accumulated evidence is no longer relevant for the task. Finally, *Figure 7—figure supplement 1i* shows the readout alignment only for time steps after stimulus inputs are passed to the network, since the accumulated evidence is always zero prior to said inputs being passed to the network and thus the accumulated evidence variational direction is ill-defined.

*Figure 7—figure supplement 1j* simply shows a closeup of the continuous integration hidden activity shown in *Figure 7j*.

## Acknowledgements

We thank Kayvon Daie, Lukasz Kusmierz, Niru Maheswaranathan, Aaron Milstein, Danil Tyulmankov for feedback on this paper. Stefan Mihalas has been in part supported by NIH grants RF1DA055669 and R01EB029813. We also wish to thank the Allen Institute for Brain Science founder, Paul G Allen, for his vision, encouragement, and support.

## Additional information

### Funding

| Funder | Grant reference number | Author |
|---|---|---|
| National Institutes of Health | RF1DA055669 | Stefan Mihalas |
| National Institutes of Health | R01EB029813 | Stefan Mihalas |
| Allen Institute | | Kyle Aitken Stefan Mihalas |

The funders had no role in study design, data collection and interpretation, or the decision to submit the work for publication.

### Author contributions

Kyle Aitken, Conceptualization, Data curation, Software, Formal analysis, Investigation, Visualization, Methodology, Writing - original draft, Writing - review and editing; Stefan Mihalas, Conceptualization, Supervision, Funding acquisition, Methodology, Project administration, Writing - review and editing

### Author ORCIDs

Kyle Aitken ⬡ http://orcid.org/0000-0003-0207-5885
Stefan Mihalas ⬡ http://orcid.org/0000-0002-2629-7100

### Decision letter and Author response

Decision letter https://doi.org/10.7554/eLife.83035.sa1
Author response https://doi.org/10.7554/eLife.83035.sa2

## Additional files

### Supplementary files
• MDAR checklist

### Data availability
The current manuscript is a computational study, so no data have been generated for this manuscript. Modeling code is available on GitHub at: https://github.com/kaitken17/mpn (copy archived at (*Aitken and Mihalas, 2023*).

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

## Appendix 1

### Analytical Approximations

#### Hadamard identity

To start, note the following Hadamard identity for matrices composed of outer products. Below, we often have multiplications of the form $\left(\left(\mathbf{hx}^T\right) \odot \mathbf{W}\right)\mathbf{x}'$. Writing out the elements of this product explicitly, we have

$$
\begin{aligned}
\left[\left(\left(\mathbf{hx}^T\right) \odot \mathbf{W}\right)\mathbf{x}'\right]_i &= \sum_{I=1}^d \left(\left(h_i x_I\right)W_{iI}\right)x_I', \\
&= h_i \sum_{I=1}^d x_I W_{iI} x_I', \\
&= h_i \sum_{I=1}^d W_{iI}\left(x_I x_I'\right), \\
&= \left[\mathbf{h} \odot \left(\mathbf{W}\left(\mathbf{x} \odot \mathbf{x}'\right)\right)\right]_i,
\end{aligned}
\tag{38}
$$

and thus, in matrix notation we have

$$
\left(\left(\mathbf{hx}^T\right) \odot \mathbf{W}\right)\mathbf{x}' = \mathbf{h} \odot \left(\mathbf{W}\left(\mathbf{x} \odot \mathbf{x}'\right)\right).
\tag{39}
$$

In words: we can effectively 'move the $\mathbf{x}$ through the $\mathbf{W}$'.

Approximation for $\mathbf{M}_t$.

We begin by deriving an expression for $\mathbf{M}_t$ in terms of all previous inputs, $\mathbf{x}_s$ for $s = 1, \ldots, t$, and the initial state $\mathbf{M}_0$. Explicitly expanding the expression from our update equations, we have

$$
\begin{aligned}
\mathbf{M}_t &= \lambda\mathbf{M}_{t-1} + \eta\mathbf{h}_t\mathbf{x}_t^T, \\
&= \lambda\mathbf{M}_{t-1} + \eta\phi\left(\left(\mathbf{M}_{t-1} \odot \mathbf{W}_{\text{inp}}\right)\mathbf{x}_t + \mathbf{W}_{\text{inp}}\mathbf{x}_t + \mathbf{b}\right)\mathbf{x}_t^T, \\
&= \lambda\left(\lambda\mathbf{M}_{t-2} + \eta\mathbf{h}_{t-1}\mathbf{x}_{t-1}^T\right) + \eta\phi\left(\left[\left(\lambda\mathbf{M}_{t-2} + \eta\mathbf{h}_{t-1}\mathbf{x}_{t-1}^T\right) \odot \mathbf{W}_{\text{inp}}\right]\mathbf{x}_t + \mathbf{W}_{\text{inp}}\mathbf{x}_t + \mathbf{b}\right)\mathbf{x}_t^T, \\
&= \lambda^2\mathbf{M}_{t-2} + \eta\lambda\phi\left(\left(\mathbf{M}_{t-2} \odot \mathbf{W}_{\text{inp}}\right)\mathbf{x}_{t-1} + \mathbf{W}_{\text{inp}}\mathbf{x}_{t-1} + \mathbf{b}\right)\mathbf{x}_{t-1}^T \\
&\quad + \eta\phi\left(\lambda\left(\mathbf{M}_{t-2} \odot \mathbf{W}_{\text{inp}}\right)\mathbf{x}_t + \eta\mathbf{h}_{t-1} \odot \mathbf{W}_{\text{inp}}\left(\mathbf{x}_{t-1} \odot \mathbf{x}_t\right) + \mathbf{W}_{\text{inp}}\mathbf{x}_t + \mathbf{b}\right)\mathbf{x}_t^T,
\end{aligned}
\tag{40}
$$

where, in reaching the last line, we have used the above Hadamard identity, *Equation 39*. So far, this expression is exact, but is clearly becoming a mess quite quickly. We can see additional steps backward in time will make this expression even more complicated, so we look for some approximation.

After training the MPN, we find $\eta = \mathcal{O}\left(1\right)$ and $\lambda = \lambda^{\max} = \mathcal{O}\left(1\right)$. Additionally, $\|\mathbf{W}_{\text{inp}}\mathbf{x}\|_2 \approx \|\mathbf{x}\|_2$, so passing the input vectors through the input layer does not significantly change their magnitude. In the last line of *Equation 40*, note that we have a term of the form $\left(\mathbf{x} \odot \mathbf{x}'\right)$. As noted in the main text, for $d \gg 1$, these terms are small relative to terms with just $\mathbf{x}$, see *Equation 8*. Additionally, this term also has a Hadamard product in the hidden activity space. If we drop the $\left(\mathbf{x}_{t-1} \odot \mathbf{x}_t\right)$ term since we know it will be small, and continue on this way, we arrive at

$$
\begin{aligned}
\mathbf{M}_t &\approx \lambda^2\mathbf{M}_{t-2} + \eta\lambda\phi\left(\left(\mathbf{M}_{t-2} \odot \mathbf{W}_{\text{inp}}\right)\mathbf{x}_{t-1} + \mathbf{W}_{\text{inp}}\mathbf{x}_{t-1} + \mathbf{b}\right)\mathbf{x}_{t-1}^T \\
&\quad + \eta\phi\left(\lambda\left(\mathbf{M}_{t-2} \odot \mathbf{W}_{\text{inp}}\right)\mathbf{x}_t + \mathbf{W}_{\text{inp}}\mathbf{x}_t + \mathbf{b}\right)\mathbf{x}_t^T, \\
&\approx \lambda^3\mathbf{M}_{t-3} + \eta\lambda^2\phi\left(\left(\mathbf{M}_{t-3} \odot \mathbf{W}_{\text{inp}}\right)\mathbf{x}_{t-2} + \mathbf{W}_{\text{inp}}\mathbf{x}_{t-2} + \mathbf{b}\right)\mathbf{x}_{t-2}^T + \\
&\quad + \eta\lambda\phi\left(\lambda\left(\mathbf{M}_{t-3} \odot \mathbf{W}_{\text{inp}}\right)\mathbf{x}_{t-1} + \mathbf{W}_{\text{inp}}\mathbf{x}_{t-1} + \mathbf{b}\right)\mathbf{x}_{t-1}^T \\
&\quad + \eta\phi\left(\lambda^2\left(\mathbf{M}_{t-3} \odot \mathbf{W}_{\text{inp}}\right)\mathbf{x}_t + \mathbf{W}_{\text{inp}}\mathbf{x}_t + \mathbf{b}\right)\mathbf{x}_t^T, \\
&\approx \lambda^t\mathbf{M}_0 + \eta\sum_{s=0}^{t-1}\lambda^s\phi\left(\lambda^{t-s}\left[\mathbf{M}_0 \odot \mathbf{W}_{\text{inp}}\right]\mathbf{x}_{t-s} + \mathbf{W}_{\text{inp}}\mathbf{x}_{t-s} + \mathbf{b}\right)\mathbf{x}_{t-s}^T
\end{aligned}
\tag{41}
$$

where in the each successive line we have dropped any $\mathbf{x} \odot \mathbf{x}'$ terms. Finally, taking $\mathbf{M}_0 = \mathbf{0}$, we arrive at the expression in the main text.

Approximation for $\mathbf{h}_t$.

Again our goal is to write $\mathbf{h}_t$ in terms of all previous inputs

$$
\begin{aligned}
\mathbf{h}_t &= \phi\left(\left(\mathbf{M}_{t-1} \odot \mathbf{W}_{\text{inp}}\right) \mathbf{x}_t + \mathbf{W}_{\text{inp}}\mathbf{x}_t + \mathbf{b}\right), \\
&= \phi\left(\left[\left(\lambda^{t-1}\mathbf{M}_0 + \eta \sum_{s=1}^{t-1} \lambda^{s-1}\mathbf{h}_{t-s}\mathbf{x}_{t-s}^T\right) \odot \mathbf{W}_{\text{inp}}\right] \mathbf{x}_t + \mathbf{W}_{\text{inp}}\mathbf{x}_t + \mathbf{b}\right), \\
&= \phi\left(\lambda^{t-1}\left[\mathbf{M}_0 \odot \mathbf{W}_{\text{inp}}\right]\mathbf{x}_t + \left[\eta \sum_{s=1}^{t-1} \lambda^{s-1}\mathbf{h}_{t-s} \odot \left(\mathbf{W}_{\text{inp}}\left(\mathbf{x}_{t-s} \odot \mathbf{x}_t\right)\right)\right] + \mathbf{W}_{\text{inp}}\mathbf{x}_t + \mathbf{b}\right),
\end{aligned}
\tag{42}
$$

where in the second line we have used the expansion for $\mathbf{M}_t$, *Equation 19*, and in the final line we have used *Equation 39*. Again, note that we have many terms that contain a $\mathbf{x} \odot \mathbf{x}'$ factor and thus are small relative to any terms with just $\mathbf{x}$. All $\mathbf{h}_{t-s}$ dependence comes with a $\mathbf{x} \odot \mathbf{x}'$ factor, and thus is already a sub-leading dependence. The leading-order approximation for a generic $\mathbf{h}_{t-s}$ is then

$$
\begin{aligned}
\mathbf{h}_{t-s} &\approx \phi\left(\lambda^{t-s-1}\left(\mathbf{M}_0\mathbf{W}_{\text{inp}}\right)\mathbf{x}_{t-s} + \mathbf{W}_{\text{inp}}\mathbf{x}_t + \mathbf{b}\right), \\
&= \phi\left(\mathbf{W}_{\text{inp}}\mathbf{x}_t + \mathbf{b}\right),
\end{aligned}
\tag{43}
$$

where in the second line we have assumed $\mathbf{M}_0 = \mathbf{0}$. Inserting this approximation into *Equation 42* and taking $\mathbf{M}_0 = \mathbf{0}$, we have

$$
\mathbf{h}_t \approx \phi\left(\left[\eta \sum_{s=1}^{t-1} \lambda^{s-1}\phi\left(\mathbf{W}_{\text{inp}}\mathbf{x}_t + \mathbf{b}\right) \odot \left(\mathbf{W}_{\text{inp}}\left(\mathbf{x}_{t-s} \odot \mathbf{x}_t\right)\right)\right] + \mathbf{W}_{\text{inp}}\mathbf{x}_t + \mathbf{b}\right).
\tag{44}
$$

Note the leading-order contribution to a given $\mathbf{h}_t$ is only dependent upon the current input, $\mathbf{x}_t$. Any dependence on previous inputs only comes in at sub-leading order.

Above, our arguments relied on the fact that the weight modulations are small as a result of the Hadamard product between vectors with components $< 1$. One could ask if the same dynamics are observed if the vector components were no longer $< 1$. To test this, we trained the network with inputs such that $\|\mathbf{x}\|_2^2 = \mathcal{O}\left(d\right)$, i.e. input vectors with components of size $\mathcal{O}\left(1\right)$. In this setting, we again found that the network found a solution such that terms from weight modulations were small relative to unmodulated contributions. This again resulted in hidden activity dynamics that were primarily driven by the most recent input to the network. We leave a further analysis of MPNs trained in settings that might increase the relative size of weight modulations for future work.

