## [Editor Report]

The study shows that fast and transient modifications of the synaptic efficacies, alone, can support the storage and processing of information over time. Convincing evidence is provided by showing that feed-forward networks, when equipped with such short-term synaptic modulations, perform a wide variety of tasks at a performance level comparable with that of recurrent networks. The results of the study are valuable to both neuroscientists and researchers in machine learning.

---

## [Decision Letter]

**Decision letter after peer review:**

Thank you for submitting your article "Neural Population Dynamics of Computing with Synaptic Modulations" for consideration by *eLife*. Your article has been reviewed by 3 peer reviewers, one of whom is a member of our Board of Reviewing Editors, and the evaluation has been overseen by Joshua Gold as the Senior Editor. The following individual involved in the review of your submission has agreed to reveal their identity: Omri Barak (Reviewer #3).

Essential revisions:

The key points consistently raised by all the reviewers in their reports are the following:

1) The main motivation of the study is unclear. Is the study proposing a specific synaptic plasticity mechanism or the goal is to expose the computational advantage(s) of transient, slow-decaying synaptic modulation? In either case, the modeling choices should be motivated more explicitly and, where possible, the link with the existing experimental observations should be made more precise.

2) It is important to test the network's performance on at least one non-integration task.

3) The most biologically-implausible aspects of the synaptic rule, such as the unbounded growth of the amplitude of the synaptic modulations, should be removed. In this context, it would be useful to relate the synaptic plasticity rule to the biology more quantitatively, at least in terms of the underlying time scales and of the required magnitude of the changes in the efficacies.

4) The study should be better contextualized in the existing literature, in order to highlight the elements of novelty in the present approach and in the results.

*Reviewer #1 (Recommendations for the authors):*

1) The Introductory paragraphs should be reworked to clarify what sort of plasticity mechanisms are being modeled.

2) Short-term plasticity mechanisms are often non-Hebbian, depending only on presynaptic activity. If I am correct that the specific plasticity rule used is understood to be a short-term Hebbian mechanism, for the authors' conclusions is it necessary that the short-term plasticity mechanism be Hebbian, or would more traditional STP mechanisms work?

3) Pg. 4, "where λ and η are parameters learned over training". This adds an element of metaplasticity, in that one typically does not think that such synaptic hyperparameters are optimized during learning a task. Does this make the network less biologically realistic? Is the selection of these during training meant simply as a route to find effective hyperparameters (that apply across tasks) but not necessarily to imply that they are actually learned? And finally, is it even necessary to learn these parameters? The reservoir computing argument in Figure 6 suggests not, though in that case the input weights are not learned either. And if learning these parameters is important for good performance on single tasks then probably not completely accurate to call M unsupervised.

4) I wondered about the robustness of the mechanism by which the MPN performs the task-the readout is based on the variation within the Go cluster and thus seems like it needs to distinguish smaller differences. Does this also make the network more sensitive to noise?

5) Related to the above, the MPN state grows quite dramatically with time ("several orders of magnitude"). Does this growth affect the robustness of the readout and does the mechanism work if the short-term plasticity saturates (and saturation seems reasonable given that synaptic efficacies likely cannot change by several orders of magnitude especially on short timescales)? How important is this large dynamic range to the network capacity?

*Reviewer #2 (Recommendations for the authors):*

The paper is well-written and the figures are carefully chosen and informative.

As I mentioned in my Public Review, one would like to understand what are the non-obvious limitations (if any) of the general principle, that is, any transient, stimulus-dependent modification of neural/synaptic properties is a memory buffer. For instance, what happens if instead of using Equation (2), you use short-term synaptic plasticity a la Tsodyks-Markram with heterogeneous parameters (e.g., different U, tau_D, and tau_F), and you only learn the synaptic weights (STP parameters are fixed)? It seems to me that this will help clarify some interesting issues, such as, do you really need the synaptic modulation to be associative (i.e., to jointly depend on the pre- and post-synaptic activity)?

*Reviewer #3 (Recommendations for the authors):*

The paper could be strengthened by tasks that are not integration-based. This could be in the context of RNNs, but then it should be clear what is the added benefit of short-term plasticity.

---

## [Author Response]

Essential revisions:The key points consistently raised by all the reviewers in their reports are the following:1) The main motivation of the study is unclear. Is the study proposing a specific synaptic plasticity mechanism or the goal is to expose the computational advantage(s) of transient, slow-decaying synaptic modulation? In either case, the modeling choices should be motivated more explicitly and, where possible, the link with the existing experimental observations should be made more precise.

The goal of this work is to explore how well networks with transient slowly-decaying synaptic modulation perform, and if they compute in a similar manner to RNNs. One of our main findings is that networks with synaptic modulation compute in a qualitatively different manner than networks with recurrence. We have modified both the abstract and introduction to better highlight this goal. Since several reviewers commented on the potential biological relevance for modulation that is only presynaptic dependent, characteristic of short-term synaptic plasticity (STSP), we have extended our results to include a modulation mechanism that is only presynaptic dependent as well (see below for more details).

Biologically, the MPN's associative modulations represent a general synapse-specific change in strength that occurs at faster relative time scales to some slow weight adjustment that in the MPN is represented by training/backpropagation. Although we do not aim to model one specific biological mechanism, we have added an additional paragraph to the Discussion section describing an example where the MPNs various plasticities could potentially map to early/late phase LTP. Similarly, the newly added presynaptic-dependent modulations could represent STSP and the slow underlying weight changes could come from LTP/LTD.

2) It is important to test the network's performance on at least one non-integration task.

We have trained the MPN on 16 additional supervised learning tasks that are relevant to neuroscience and added these results to the main text (Sec. 3.3, Figure 7). A few examples are: contextual decision making, delay match sample, interval discrimination, motor timing, etc. This was done using the NeuroGym package (Molano-Mazon et al., 2022). Across all tasks we investigate, the MPN achieves comparable performance to its RNN counterparts and even outperforms them on certain tasks.

3) The most biologically-implausible aspects of the synaptic rule, such as the unbounded growth of the amplitude of the synaptic modulations, should be removed. In this context, it would be useful to relate the synaptic plasticity rule to the biology more quantitatively, at least in terms of the underlying time scales and of the required magnitude of the changes in the efficacies.

We note that although the modulations are not explicitly bounded, the constant decay of modulations effectively limits their size (see saturation for λ < 1 in Figure 4c). Also note the saturation did not seem to affect the accuracy of the networks considerably (Figure 4d). Additionally, we found that effects of the modulations on the hidden activity were generally small compared to that of the fixed weights (see Equation (20), which yields analytical approximations shown in Figure 5c and 5d).

However, to be complete, we have now also tested how adding explicit bounds to the modulations affected the MPN’s performance and behavior. We have added these results to a footnote in the main text, additional figures in the supplement, and a short discussion in the methods section. In short, unless bounds are prohibitively restrictive (beyond what we believe is biologically reasonable), we do not find a significant change in either the performance or behavior of the network. For example, limiting the modulations to only change by 10% of the weight’s original value (quite a bit more restrictive than experiment, see below), we found only a few percentage drop in accuracy on integration tasks. When we introduced explicit bounds, we found that the network adapted its learning rate to compensate for the loss of information that could occur when a modulation could no longer be changed. That is, more restrictive bounds on modulations caused the network to use smaller learning rates (i.e. smaller eta) so that it did not run into the modulation bounds as quickly.

Biologically, STDP that potentiates synapses up to ~200% of their original values and depresses by up to ~30% of original values at timescales of 10s of minutes has been seen in a wide variety of studies in various parts of the brain (Sjöström and Häusser, 2006; McFarlan et al., 2022). Other specific studies sometimes find values that are a bit more extreme, for example depression as low as 20% in Cho, K. et al., 2000. Over shorter time scales, STSP has been shown to facilitate to a similar relative magnitude and seems to be capable of almost complete depression (Campagnola et al., 2022).

McFarlan, Amanda R., et al. "The plasticitome of cortical interneurons." Nature Reviews Neuroscience (2022): 1-18.

Sjöström PJ, Häusser M. “A cooperative switch determines the sign of synaptic plasticity in distal dendrites of neocortical pyramidal neurons.” Neuron 51: 227–238, 2006.

Cho, K. et al. “A new form of long-term depression in the perirhinal cortex.” Nat. Neurosci. 3, 150–156, 2000.

4) The study should be better contextualized in the existing literature, in order to highlight the elements of novelty in the present approach and in the results.

We have added several comparisons to references suggested by the reviewers in the related work and Discussion sections as well as throughout the work where relevant.

As mentioned above, to address several reviewers’ inquiry into whether an associative mechanism was necessary (and to extend this work to model STSP-like mechanisms), we have introduced and analyzed a special case of the modulation updates that is only presynaptic dependent. That is, it is the same modulation rule as the MPN, except the hidden state dependence has been removed. We test this new modulation setting in an otherwise identical network/training setup as the MPN. We call this network the “MPNpre” and have added results throughout our paper analyzing the performance and behavior of this non-associative mechanism relative to the MPN. In short, due to the similarity of their operation, the MPNpre dynamics are largely the same as the MPN – dynamics are driven foremost by the current input and information about the history of the phrase is stored in the modulation. Additionally, for the majority of tasks we consider in this work (including the 16 additional tasks), the MPNpre performs at levels comparable to the MPN, often only slightly worse due to its simpler updates.

Reviewer #1 (Recommendations for the authors):1) The Introductory paragraphs should be reworked to clarify what sort of plasticity mechanisms are being modeled.

See our response to your public review.

2) Short-term plasticity mechanisms are often non-Hebbian, depending only on presynaptic activity. If I am correct that the specific plasticity rule used is understood to be a short-term Hebbian mechanism, for the authors' conclusions is it necessary that the short-term plasticity mechanism be Hebbian, or would more traditional STP mechanisms work?

We have added several results to our work for a modulation update rule only dependent upon presynaptic firing rates, see our response to Essential Revisions for more details.

3) Pg. 4, "where λ and η are parameters learned over training". This adds an element of metaplasticity, in that one typically does not think that such synaptic hyperparameters are optimized during learning a task. Does this make the network less biologically realistic? Is the selection of these during training meant simply as a route to find effective hyperparameters (that apply across tasks) but not necessarily to imply that they are actually learned? And finally, is it even necessary to learn these parameters? The reservoir computing argument in Figure 6 suggests not, though in that case the input weights are not learned either. And if learning these parameters is important for good performance on single tasks then probably not completely accurate to call M unsupervised.

Indeed, this adds an element of meta-learning to the network setup. The ability for eta and λ to be adjustable mostly came from a desire to allow for a flexible learning rule and see if there were trends to what the network preferred in order to learn certain computations. After training the MPN across several types of tasks, we learned that allowing the network to adjust said parameters during learning was not absolutely necessary. In practice, we found that λ always saturated its upper bound (so that it retained information for as long as possible) and the adjustment of the rest of the weights in the network could compensate for choosing different fixed eta (both for magnitude and sign). For instance, for smaller eta, the smaller relative effects of modulations on the hidden activity could be magnified by the network learning larger readouts.

4) I wondered about the robustness of the mechanism by which the MPN performs the task-the readout is based on the variation within the Go cluster and thus seems like it needs to distinguish smaller differences. Does this also make the network more sensitive to noise?

We did find that the MPN was slightly less robust to noise than the RNN counterparts (Figure 6c), but we more attributed it to the lack of a task-dependent attractor structure that provides some robustness to perturbations away from the attractor. This difference in perturbation behavior would certainly manifest in the hidden activity of both networks. Note that the output neuron values are dependent upon both the readout vector size and the size of variations of the hidden activity, so although the encoding of the MPN’s accumulated evidence may vary more rapidly in hidden activity space than its RNN counterparts, it could be compensated with larger readouts. Notably, the network performs well even in the new tests we ran where modulation size was severely restricted, making the variation due to accumulated evidence even smaller (Figure 3-figure supplement 2h).

5) Related to the above, the MPN state grows quite dramatically with time ("several orders of magnitude"). Does this growth affect the robustness of the readout and does the mechanism work if the short-term plasticity saturates (and saturation seems reasonable given that synaptic efficacies likely cannot change by several orders of magnitude especially on short timescales)? How important is this large dynamic range to the network capacity?

This was a big question we had as well, and we believe the plots in Figures 4c and 4d support the fact that the saturation of the SM matrix does not seem to affect the network’s ability to store information significantly. Note that for λ < 1, the final state magnitude begins to saturate (Figure 4c), yet the accuracies do not drop significantly once it has saturated (Figure 4d).

We also tested this more explicitly by introducing bounds on the modulation growth size and found that the network continued to perform quite well (see response to Essential Revisions above).

Reviewer #2 (Recommendations for the authors):The paper is well-written and the figures are carefully chosen and informative.As I mentioned in my Public Review, one would like to understand what are the non-obvious limitations (if any) of the general principle, that is, any transient, stimulus-dependent modification of neural/synaptic properties is a memory buffer. For instance, what happens if instead of using Equation (2), you use short-term synaptic plasticity a la Tsodyks-Markram with heterogeneous parameters (e.g., different U, tau_D, and tau_F), and you only learn the synaptic weights (STP parameters are fixed)? It seems to me that this will help clarify some interesting issues, such as, do you really need the synaptic modulation to be associative (i.e., to jointly depend on the pre- and post-synaptic activity)?

As mentioned in our public response, we explored a pre-synaptic only rule and found that it did not change the results significantly. It continues to perform well in the reservoir computing setting, and so indeed the training of the eta and λ parameters is not strictly necessary for it to perform well if they are set to reasonable values.

Reviewer #3 (Recommendations for the authors):The paper could be strengthened by tasks that are not integration-based. This could be in the context of RNNs, but then it should be clear what is the added benefit of short-term plasticity.

See our response to your public review.